# Nonlinear transcriptional responses to gradual modulation of transcription factor dosage

**Júlia Domingo[1][*][†], Mariia Minaeva[2][‡], John A Morris[1,3], Samuel Ghatan[1], Marcello Ziosi[1], Neville E Sanjana[1,3], Tuuli Lappalainen[1,2]***

[1]New York Genome Center, New York, United States; [2]Science for Life Laboratory, Department of Gene Technology, KTH Royal Institute of Technology, Stockholm, Sweden; [3]Department of Biology, New York University, New York, United States

**\*For correspondence:**
julia.domingo.espinos@gmail.
com (JD);
tlappalainen@nygenome.org (TL)

**Present address:** [†]Allostery
Exploration Technologies,
Barcelona, Spain; [‡]Institute
of Computational Biology,
Helmholtz Center, Munich,
Germany

## eLife Assessment

This **important** work develops a new protocol to experimentally perturb target genes across a quantitative range of expression levels in cell lines. The evidence supporting their new perturbation approach is **convincing**, and we propose that focusing on single modality (activation or inhibition) would be sufficient to draw their conclusions. The study will be of broad interest to scientists in the fields of functional genomics and biotechnology.

**Abstract** Genomic loci associated with common traits and diseases are typically non-coding and likely impact gene expression, sometimes coinciding with rare loss-of-function variants in the target gene. However, our understanding of how gradual changes in gene dosage affect molecular, cellular, and organismal traits is currently limited. To address this gap, we induced gradual changes in gene expression of four genes using CRISPR activation and inactivation in human-derived K562 cells. Downstream transcriptional consequences of dosage modulation of three master trans-regulators associated with blood cell traits (*GFI1B*, *NFE2*, and *MYB*) were examined using targeted single-cell multimodal sequencing. We showed that guide tiling around the TSS is the most effective way to modulate *cis* gene expression across a wide range of fold changes, with further effects from chromatin accessibility and histone marks that differ between the inhibition and activation systems. Our single-cell data allowed us to precisely detect subtle to large gene expression changes in dozens of *trans* genes, revealing that many responses to dosage changes of these three TFs are nonlinear, including non-monotonic behaviours, even when constraining the fold changes of the master regulators to a copy number gain or loss. We found that the dosage properties are linked to gene constraint and that some of these nonlinear responses are enriched for disease and GWAS genes. Overall, our study provides a straightforward and scalable method to precisely modulate gene expression and gain insights into its downstream consequences at high resolution.

## Introduction

Precision control of gene expression levels plays a pivotal role in defining cell type specificity and coordinating responses to external stimuli. Imbalances in this intricate regulation can underlie the genetic basis of both common and rare human diseases. The vast majority of genetic variants associated with complex disease, as revealed by genome-wide association studies (GWAS), are located in noncoding regions, with likely gene regulatory effects (*Maurano et al., 2012*). Previous studies have attempted to elucidate these effects by mapping genetic associations to gene expression (*Aguet*

*et al., 2020*; *Claussnitzer et al., 2020*), and more recently, CRISPR-based perturbations of GWAS loci have provided insights into their functional consequences (*Morris et al., 2023*). A major driver of rare genetic diseases is loss-of-function variants affecting one or both copies of the gene, leading to disease via dramatic reduction of functional gene dosage (*Zschocke et al., 2023*). The substantial overlap (*Backman et al., 2021*; *Freund et al., 2018*) and potential joint effects (*Castel et al., 2018*; *Fahed et al., 2020*) of rare and common variants indicate a general link between different degrees of perturbation of gene dosage and disease phenotypes.

However, our understanding of the quantitative relationship between gradual changes in gene dosage and downstream phenotypes remains elusive for most human genes. Practical applications of the compelling allelic series concept to identify genes where increasingly deleterious mutations have increasing phenotypic effects have been limited by the sparsity of segregating variants with an impact on a given gene in the human population (*McCaw et al., 2023*). Experimental characterization of gene function in model systems has predominantly relied on gene knock-out or knock-down approaches (*Sanjana, 2017*). While these studies have proven useful to identify dosage-sensitive genes involved in cellular functions and disease (*Collins et al., 2022*; *Fehrmann et al., 2015*; *Wang et al., 2015*; *Hart et al., 2015*; *Cowley et al., 2014*), these approaches only provide a limited discrete relationship between the number of functional gene copies and a certain phenotype (e.g. loss-of-function consequence vs. wild-type). However, such relationships are in fact determined by continuous dosage-to-phenotypes functions that, as suggested by a small number of previous experimental studies (*Keren et al., 2016*; *Jost et al., 2020*; *Hawkins et al., 2020*), can be complex and thus are challenging to infer from loss-/gain-of-function data.

Recently, new methods have enabled the gradual modulation of gene dosage in model systems (*Jost et al., 2020*; *Noviello et al., 2023*; *Chiarella et al., 2020*; *Liu et al., 2024*), while large-scale insights into the downstream effects of dosage modulation have largely come from yeast (*Keren et al., 2016*) and bacteria (*Hawkins et al., 2020*; *Lalanne et al., 2021*), demonstrating that nonlinear relationships between gene dosage and phenotype are common. In humans, the relationship between dosage and downstream phenotypes is largely unexplored. Only a few limited studies (*Keren et al., 2016*; *Jost et al., 2020*; *Hawkins et al., 2020*) have dissected these consequences. For instance, the disease-associated transcription factor *SOX9* (*Naqvi et al., 2023*) showed a nonlinear relationship between dosage and multiple tiers of phenotypes, including DNA accessibility, RNA expression of downstream targets, raising the question of whether this phenomenon occurs with other transcription factors. More recently, similar evidence has been shown in the case of the *NKX2-1* lineage factor with an oncogenic role in lung adenocarcinoma (*Pulice and Meyerson, 2023*). Generally, transcription factors represent a particularly compelling target for the characterization of gene dosage effects. They are key regulators of cellular functions, enriched for disease associations (*Mostafavi et al., 2023*) and often classified as haploinsufficient (*van der Lee et al., 2020*). Additionally, their effects can be measured by transcriptome analysis. However, our knowledge of their dosage-dependent effects on regulatory networks still remains limited, particularly regarding subtle dosage variation within their natural range (*Liu et al., 2024*).

In this study, we developed and characterized a scalable novel sgRNA design approach for gradually decreasing and increasing gene dosage with the CRISPR interference (CRISPRi) and activation (CRISPRa) systems. We applied this to four genes, with single-cell RNA sequencing (scRNA-seq) as a cellular readout of downstream effects. While classic Perturb-Seq analyses have focused on gene knockdown effects, we assess the effects of gradual up- and down-regulation of target genes. We uncovered quantitative patterns of how gradual changes in transcription dosage lead to linear and nonlinear responses in downstream genes. Many downstream genes are associated with rare and complex diseases, with potential effects on cellular phenotypes.

## Results
### Precise modulation and quantification of gene dosage using CRISPR and targeted multimodal single-cell sequencing

We selected four genes for gradual modulation of their dosage in the human erythroid progenitor cell line K562 (*Ulirsch et al., 2016*): *GFI1B*, *NFE2*, *MYB*, and *TET2*. Two of the genes, *GFI1B* and *NFE2*, have been implicated in blood diseases and traits (*Möröy et al., 2015*; *Polfus et al., 2016*; *Jutzi*

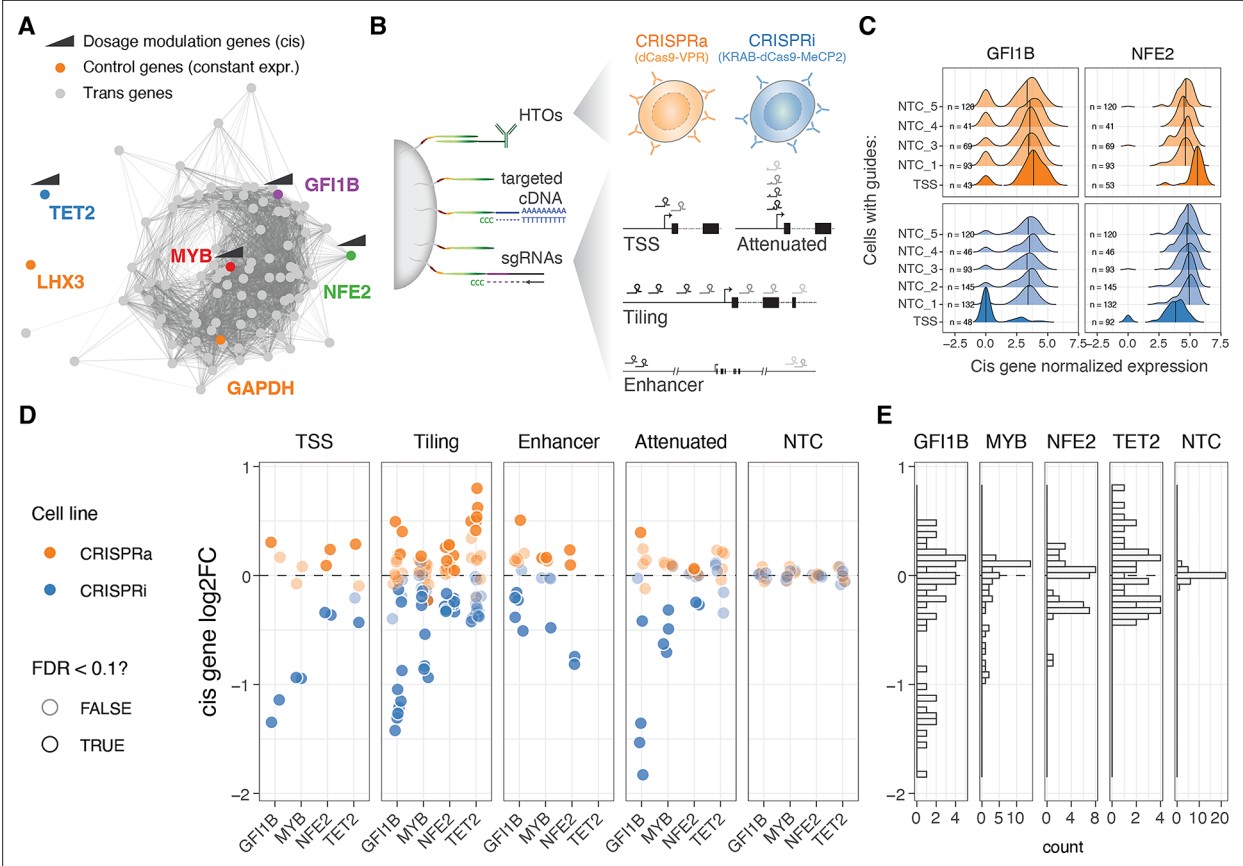

**Figure 1.** Modulation and quantification of gene dosage using CRISPR and targeted multimodal single-cell sequencing. (**A**) Co-expression network representation of the 92 selected genes under study. Genes (nodes) are connected by edges when their co-expression across single cells was above 0.5 (data used from *Morris et al., 2023*). Highlighted in colour are the two control highly (GAPDH) and lowly (LHX3) constantly expressed genes, as well as cis genes for which dosage was modulated with CRISPRi/a. (**B**) Design of the multimodal single-cell experiment (HTO = hash tag oligos). (**C**) Distribution of the GFI1B (left) or NFE2 (right) normalized expression across single cells for different classes of sgRNAs (NTC = Non-targeting controls, TSS = transcription start site). (**D**) Resulting relative expression change (log2 fold change) of the 4 cis genes upon each unique CRISPR perturbation when grouped across different classes of sgRNAs. (**E**) Distribution of cis gene log2FC across all sgRNA perturbations.

The online version of this article includes the following figure supplement(s) for figure 1:

**Figure supplement 1.** Experimental design and data processing from UMIs to expression fold change, related to *Figure 1* and STAR methods.

**Figure supplement 2.** Biochemical and activity properties of different types of single guide RNAs (sgRNAs).

**Figure supplement 3.** Gradual effects of the single guide RNAs (sgRNAs).

*et al., 2019*), and in our earlier work, we identified a broad transcriptional response to inhibition of GWAS-overlapping enhancers to these genes (*Morris et al., 2023*). *MYB* is a key transcription factor (*Baker et al., 2014*) and a downstream target of *GFI1B* (*Morris et al., 2023*). *TET2* has a role in DNA demethylation and is unrelated to these transcriptional networks and is included in this study as a control with minimal expected *trans* effects. We refer to these four genes, targeted in cis for modulation of their regulation, as *cis* genes (*Figure 1A*).

To modulate the gene expression of the *cis* genes, we use K562 cells expressing CRISPRi (KRAB-dCas9-MECP2) and CRISPRa (dCas9-VPR) systems (see Methods). Both cell lines were hashed with DNA-conjugated antibodies against different surface proteins that allow pooled experiments. To obtain a wide range of dosage effects, we used four different single guide RNA (sgRNA) design strategies (*Figure 1B*): (1) targeting the transcription start site (TSS) as in the standard CRISPRi/CRISPRa approach, (2) tiling sgRNAs +/-1000 bp from the TSS in approximately 100 bp intervals, (3) targeting known *cis*-regulatory elements (CREs), and (4) using attenuated guides that target the TSS but contain mismatches to modulate their activity (*Jost et al., 2020*). We further included five non-targeting control (NTC) sgRNAs as negative controls.

The library of altogether 96 guides was transduced to a pool of K562-CRISPRi and K562-CRISPRa cells at low multiplicity of infection (MOI). After eight days, we performed ECCITE-seq (see Methods) to capture three modalities: cDNA, sgRNAs, and surface protein hashes (oligo-tagged antibodies with unique barcodes against ubiquitously expressed surface proteins). Instead of sequencing the full transcriptome, we used target hybridization to capture a smaller fraction of the cDNA and obtain more accurate expression readouts at a feasible cost. The subset of selected transcripts were picked from the transcriptional downstream regulatory networks of *GFI1B* and *NFE2* identified previously (*Morris et al., 2023*), maintaining similar patterns of co-expression correlation across co-expression clusters (see Methods, *Figure 1—figure supplement 1A*). We targeted a total of 94 transcripts (*Figure 1A*), including the four *cis* genes, 86 genes that represent trans targets of *GFI1B* and/or *NFE2* (*Morris et al., 2023*; *Figure 1—figure supplement 1A*), *LXH3* that is not expressed in blood progenitors, *GAPDH* that is highly expressed and often considered an invariable housekeeping gene and the dCas9-VPR or KRAB-dCas9-MeCP2 transcripts.

We used the protein hashes and the dCas9 cDNA (indicating the presence or absence of the KRAB domain) to demultiplex and determine the cell line—CRISPRi or CRISPRa. Cells containing a single sgRNA were identified using a Gaussian mixture model (see Methods). Standard quality control procedures were applied to the scRNA-seq data (see Methods). To confirm that the targeted transcript capture approach worked as intended, we assessed concordance across capture lanes (*Figure 1—figure supplement 1C*). The final data set had 20,001 cells (10,647 CRISPRi and 9354 CRISPRa), with an average of 81 and 86 cells with a unique sgRNA for the CRISPRa and CRISPRi, respectively (*Figure 1—figure supplement 1D*).

## Gradual modulation of gene expression across a broad range with CRISPRi/a

Next, we calculated the expression fold change for each of the four *cis* genes targeted by each sgRNA in the two cell lines (CRISPRi/a), comparing each group of cells with its respective NTC sgRNA group (see Methods). We first confirmed that the sgRNAs targeting the transcription start site (TSS) up- and down-regulated their targets (*Figure 1C*, *Figure 1—figure supplement 1F*). When looking at all sgRNAs at once, across the four genes, we observed a 2.3-fold range (*Figure 1E*), with a minimum 72% reduction and maximum 174% increased expression (log2(FC) values from −1.83–0.80). However, the range varied between the genes, with *GFI1B* covering the widest range of gene expression changes (gene expression ranging between 0.28–1.42 fold), while *MYB* expression could not be pushed higher than 1.13 fold (*Figure 1E*). The direction of the effects were consistent with the cell lines of origin, where 98.88% of the significant perturbations (Wilcoxon rank test at 10% FDR, n=89) were correctly predicted based on the direction of the target gene fold change. The predicted on- and off-target properties of the guides (*Doench et al., 2014*; *Doench et al., 2016*; *McKenna and Shendure, 2018*) did not correlate with the fold changes in the *cis* genes (*Figure 1—figure supplement 2A*), suggesting that the observed effects represent true *cis*-regulatory changes. The fold changes were also robust to the number of cells containing a particular sgRNA (*Figure 1—figure supplement 2B*, top).

We verified that the fold change estimation was not biased depending on the expression level of the target gene at the single-cell level, which can vary due to drop-out effects or binary on/off effects of the KRAB-based CRISPRi system (*Noviello et al., 2023*). By splitting cells with the same sgRNA based on the normalized expression of the *cis* gene (0 vs. >0 normalized UMIs, *Figure 1—figure supplement 3A*), we observed highly concordant transcriptome gene expression effects between the two groups (*Figure 1—figure supplement 3B*). This indicates that the dosage changes per guide were not primarily driven by the changing frequency of binary on/off effects, and the use of pseudo-bulk fold changes provides a robust estimation of *cis* gene fold changes. These patterns are further supported by the cells forming a gradient rather than distinct clusters on a UMAP (*Figure 1—figure supplement 3C*).

The fold change patterns differed between sgRNA designs (*Figure 1D*, left). As expected, sgRNAs targeting the TSS showed strong perturbations in gene expression. However, sgRNAs tiled +/-1 kb from the TSS provided a broader and more gradual range of up- and downregulation across the target genes, sometimes surpassing the effects of TSS-targeting sgRNAs. Attenuated sgRNAs with mismatch mutations resulted in a range of gene silencing effects in the CRISPRi line, as expected based on their original design (*Jost et al., 2020*). However, these attenuated sgRNAs did not exhibit

such a dynamic range in the CRISPRa modality, although a significant correlation existed between the silencing or activating effect size and the distance of the mismatch from the protospacer adjacent motif (PAM) when considering all data points together (*Figure 1—figure supplement 2C*). The sgRNAs targeting distal *cis*-regulatory elements (CREs) showed both inhibiting and activating effects, even though both the CRISPRi and CRISPRa constructs were initially designed to inhibit or activate transcription from the promoter and initial gene body region. Nonetheless, the number of known CREs per gene is typically limited. Given its simplicity and the ability to achieve both up- and downregulation of the target gene, we consider the tiling sgRNA approach, with a simple design that only requires annotation of the TSS, as a useful method for gradually modulating gene dosage with CRISPRi/a systems.

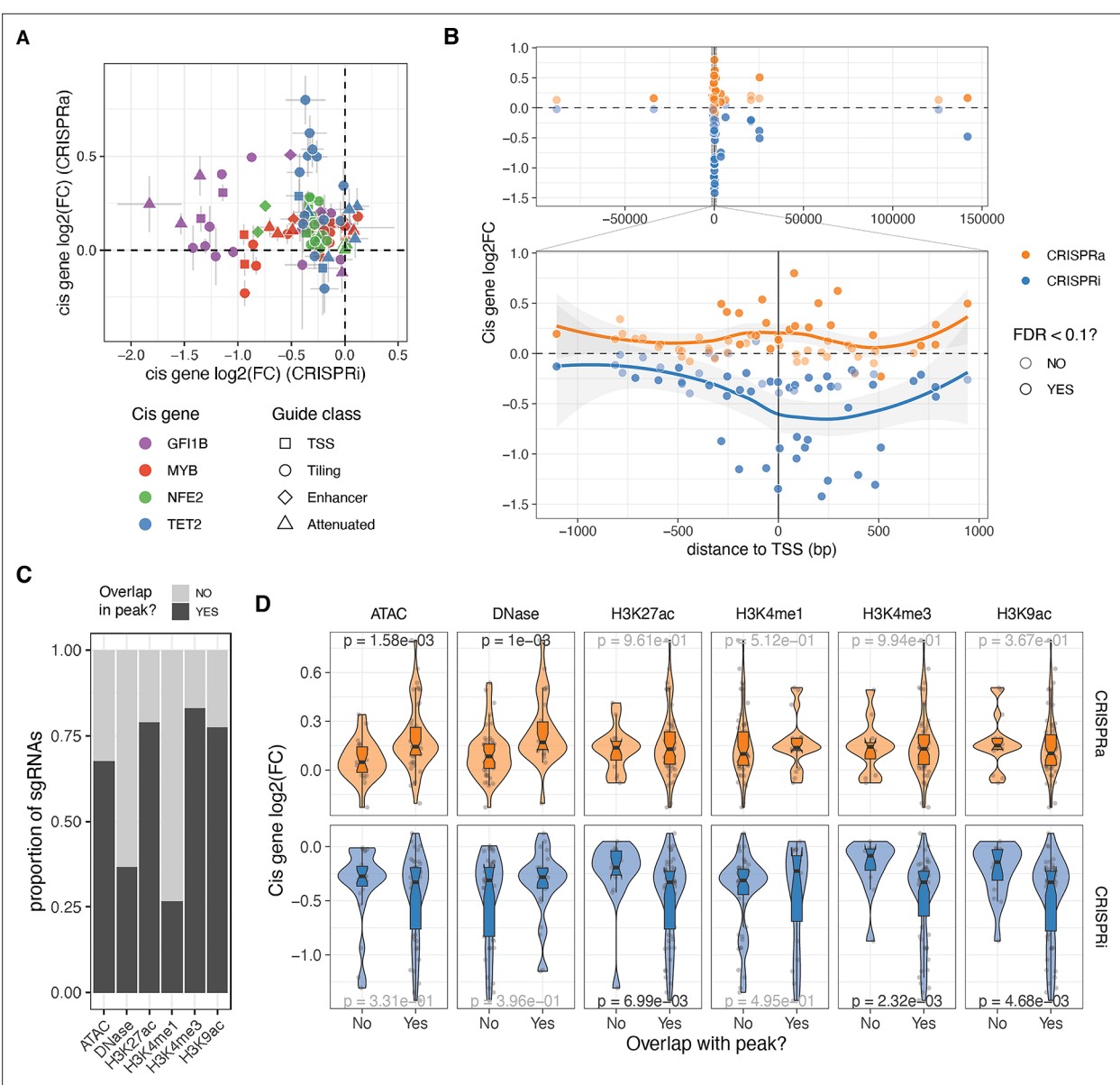

**Figure 2.** Cis determinants of dosage. (**A**) Comparison of the relative expression change (log2FC) from the same single guide RNA (sgRNA) between the two different CRISPR modalities. Vertical and horizontal bars represent CRISPRa and CRISPRi standard errors, respectively. (**B**) Relative expression change of the targeted cis gene based on distance from transcription start site (TSS). Top plot excluded attenuated and non-targeting control (NTC) sgRNAs, while bottom plot also excludes enhancer sgRNAs. (**C**) Number of sgRNAs that overlap with the different epigenetic or open chromatin peaks. (**D**) Relative expression change to NTC sgRNAs (log2(FC)) of all cis genes when their sgRNAs fall or not in the different epigenetic or open chromatin peaks. P-value results from Wilcoxon rank-sum tests, with nominally significant p-values shown in black.

## *Cis* determinants of dosage

Having designed guides targeting both distal and local neighbouring regulatory regions of the four transcription factors (TFs) and ensuring minimal bias in fold changes due to sgRNA's biochemical properties, we investigated the *cis* features that determine the strength of dosage perturbation. We observed substantial differences in the effects of the same guide on the CRISPRi and CRISPRa backgrounds, with no significant correlation between *cis* gene fold changes (*Figure 2A*). However, in both modalities, the strongest effects on gene expression were observed when the guides were close to the transcription start site (TSS) (*Figure 2B*, excluding NTC and attenuated sgRNAs), although the peaks of strongest activation or repression differed between the modalities. In the CRISPRi modality, the maximum effect was located within the gene body at +238 bp from the TSS (*Figure 2B*, bottom), consistent with previous studies that used essentiality as a proxy for expression (*Sanson et al., 2018*). However, in the CRISPRa modality, the maximum average fold changes occurred closer to the TSS at around –99 bp (*Figure 2B*, bottom), as also shown for CD45 (*Legut et al., 2020*).

Enhancer, tiling, and TSS sgRNAs targeted regions of the genome with different compositions of histone marks, annotated by ENCODE, in K562 cells (*Luo et al., 2020*; *Figure 2C*), which allowed us to investigate the impact of chromatin state on the strength of *cis* gene dosage modulation. The magnitude of *cis* gene fold changes varied significantly depending on the presence of specific marks or peaks, which again differed between the two modalities (*Figure 2D*). In the CRISPRa cell line, the strongest effects were observed when guides were located in regions with open chromatin marks, such as DNase or ATAC peaks. In contrast, the strongest repression by CRISPRi occurred in genomic regions with the presence of H3K27ac, H3K4me3, and H3K9ac marks. These differences may be explained by the distinct mechanisms of action of the activator and repressor domains. MeCP2 and KRAB repressor domains recruit additional repressors that silence gene expression through chromatin remodelling activities, such as histone deacetylation (*Lupo et al., 2013*). On the other hand, the VPR activation fusion domain may only require Cas9 to scan the open chromatin and recruit RNA polymerase and additional transcription factors to activate transcription. Overall, while a few sgRNAs have a strong effect in both CRISPRi and CRISPRa cell lines, a single guide library containing guides optimized for both modalities enables a range of gradual dosage regulation. However, larger data sets are needed for more careful modelling of the ideal dosage modulation designs and to understand how both *cis*-regulatory features, feedback loops, and other mechanisms contribute to the outcomes.

## *Trans* responses of transcription factor dosage modulation

We then turned our attention to the remaining 91 genes captured by our custom panel and determined the relative expression fold change of each *trans* gene, compared to NTC in each unique guide perturbation (see Methods). Principal component analysis (PCA) performed on all pseudo-bulk fold changes demonstrated the removal of batch effects from the cell lines and revealed a clear direction of the *cis* gene dosage effect in the first three principal components (*Figure 3—figure supplement 1B*). This finding suggests that dosage modulation is the primary determinant of *trans* effects. The PCA indicated that the dosage modulation of *GFI1B* and *MYB* is reflected in opposite directions in PC1 and PC2, while the *trans* responses of *NFE2* are captured by PC3.

Using a false discovery rate (FDR) cutoff of 0.05, all 91 *trans* genes except for the neural-specific TF *LHX3* (negative control) exhibited a significant change in expression upon perturbation of any of the TFs. The observed trans-effects were well correlated with perturbations of these genes in other data sets (*Figure 3—figure supplement 1C and D*). Among all measured fold changes, the most extreme negative effect sizes were observed in cis genes, with the top 10 being predominantly reductions in *GFI1B* expression. This indicates that *cis* downregulation tended to surpass the endogenous expression limits. In contrast, the largest increases in gene expression were observed through *trans* mechanisms, where *KLK1* and *TUBB1* reached the largest expression values when *GFI1B* was strongly upregulated, or *SPI1* and *DAPK1* when *GFI1B* was strongly downregulated. These findings suggest that the CRISPRa approach did not reach a biological ceiling of overexpression.

Inspecting *trans* responses as a function of *cis* gene modulation, we observed that the number of expressed genes and the mean absolute expression changes of *trans* genes exhibited gene-specific correlations with *cis*-gene dosage (*Figure 3A*, *Figure 3—figure supplement 1E*). Perturbations in *GFI1B* led to the most pronounced *trans* responses, with positive dosage changes resulting in larger effect sizes compared to decreasing TF gene expression, where the effect plateaued. *NFE2* exhibited

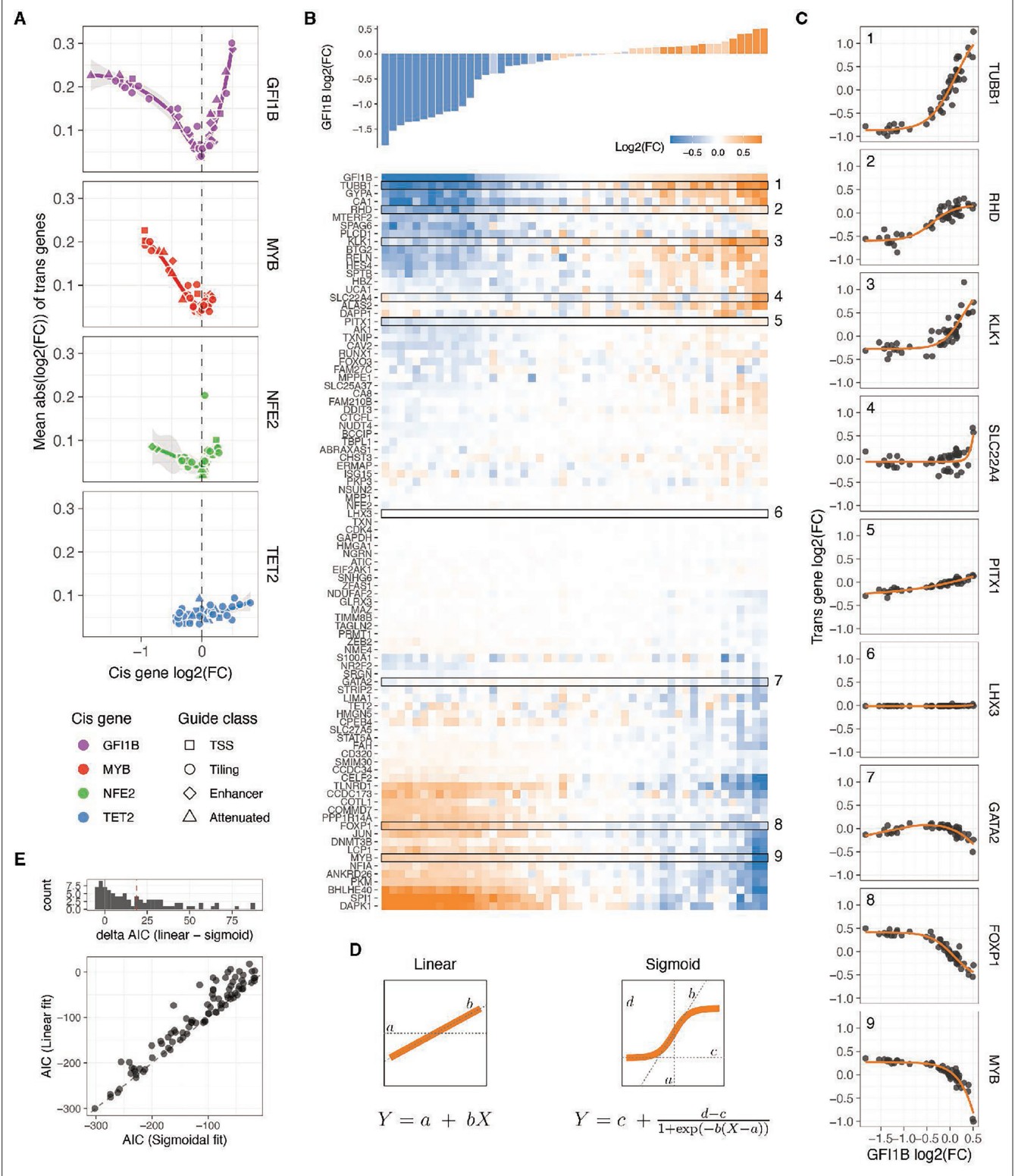

**Figure 3.** Trans responses of transcription factor dosage modulation. (**A**) Average absolute expression change of all trans genes relative to the changes in expression of the cis genes. (**B**) Changes in relative expression of all trans genes (bottom heatmap) in response to GFI1B expression changes (top barplot) upon each distinct targeted single guide RNA (sgRNA) perturbation, in comparison to non-targeting control (NTC) cells. The rows of the heatmap (trans genes) are hierarchically clustered based on their expression fold change linked to alterations in GFI1B dosage. Highlighted rows are selected dosage response examples shown in C. (**C**) Dosage response curves of the highlighted trans gene in B as a function of changes in GFI1B expression. The orange line represents the sigmoid model fit, except for GATA2, which displays a non-monotonic response and are fitted with a loess

*Figure 3 continued on next page*

*Figure 3 continued*

curve. (**D**) Illustration of the linear and sigmoid models and equations used to fit the dosage response curves. (**E**) Distribution of the difference in Akaike Information Criterion ($\Delta AIC_{linear\text{-}sigmoid}$) after fitting the sigmoidal or linear model for each trans gene upon GFI1B dosage modulation (top panel), and the direct comparison of the Akaike Information Criterion (AIC) of each fit (bottom panel).

The online version of this article includes the following figure supplement(s) for figure 3:

**Figure supplement 1.** Global view of trans effects and their replication.

**Figure supplement 2.** Trans gene responses to GFI1B dosage modulation.

**Figure supplement 3.** Trans gene responses to MYB dosage modulation.

**Figure supplement 4.** Trans gene responses to NFE2 dosage modulation.

**Figure supplement 5.** Trans gene responses to TET2 dosage modulation.

**Figure supplement 6.** Dosage response linear and non-linear model fitting.

**Figure supplement 7.** Distribution of the fitted parameters of the sigmoidal model on dosage responses.

similar patterns but with a smaller magnitude. In the case of *MYB*, *trans* responses were observed when decreasing the expression of this TF, but the effects of upregulation are largely unknown as we were unable to increase *MYB* expression beyond 0.35. As expected, given the unrelatedness of *TET2* to the *trans* network, dosage modulation of this gene had minimal *trans* effects with the least pronounced trend when compared to *TET2* dosage, so we excluded it from subsequent analyses.

## Widespread nonlinear dosage responses in *trans* regulatory networks

Upon clustering the changes in expression of *trans* genes based on the *cis* gene dosage change linked to each sgRNA, we identified distinct clusters exhibiting different dosage-response patterns (***Figure 3B*** for *GFI1B*, ***Figure 3—figure supplements 2–5A*** for all *cis* genes). Further examination of the gene expression fold changes for each individual trans gene in relation to the TF fold changes revealed a diverse range of response patterns (***Figure 3C***, ***Figure 3—figure supplements 2–5B*** for all *cis* genes). These responses exhibited both linear and nonlinear forms, including some instances of non-monotonic gene expression responses for certain *trans* genes within the *GFI1B* trans network (e.g. *GATA2* in ***Figure 3C***, ***Figure 3—figure supplement 6E***).

To accurately characterise the dosage response, we employed both linear and nonlinear modelling approaches (***Figure 3D***), which allowed us to quantitatively assess the extent of nonlinear responses by comparing the goodness of fit of these models using the Akaike Information Criterion (AIC). For the nonlinear model, we utilized a sigmoid function with four free parameters (***Figure 3D***, right). These parameters represented the slope at the inflection point (*b*, indicating the rate of increase or decrease in expression), the minimum and maximum asymptotes (*c* and *d*, representing the lower and upper limits of fold change), and the value of *cis* gene expression at which the inflection point occurs (*a*). To prevent overfitting, we implemented a 10-fold cross-validation scheme, which yielded reliable predictions on the left-out data (Pearson $r$=0.71–0.88 for all *trans* genes in the *GFI1B*, *MYB*, and *NFE2* networks, ***Figure 3—figure supplement 6C***). Additionally, the predicted parameter *a* was centred around zero, as expected, since the input data represents relative fold changes (***Figure 3—figure supplement 7***). Since a sigmoid function cannot capture non-monotonic responses, we employed a loess regression as an alternative approach for the few genes that exhibited non-monotonic responses (see Methods, ***Figure 3—figure supplement 6D and E***). For the vast majority of genes, the sigmoid (or loess) fit was remarkably good, partially due to the low level of noise in the targeted scRNA-seq data.

We compared the performance of the linear vs. nonlinear models with the $\Delta AIC$ ($AIC_{linear} - AIC_r$), where a positive $\Delta AIC$ means that the sigmoid model captures the variance better in the dosage response than in the linear model. This showed that most *GFI1B*-dependent dosage expression responses are better fit by the sigmoid model (median $\Delta AIC$ = 18.7, with 70.4% of all *trans* genes with a significant response having $\Delta AIC$ >2, ***Figure 3E***). The responses to dosage modulation of *MYB* and *NFE2* were also better captured by the nonlinearities, but to a lesser extent (0.14 and 3.4 median $\Delta AIC$, with 20.8% and 40.7% of all *trans* genes dosage responses having $\Delta AIC$ >2 for *MYB* and *NFE2*, respectively, ***Figure 3—figure supplement 6A***). The broader range of *GFI1B* expression modulation, providing more data to detect nonlinear trends, likely contributes to this difference. When ignoring those genes classified as unresponsive (genes that their expression did not change

upon the TF modulator, see Methods), even more responses of the remaining *trans* genes were better explained by a sigmoidal model with 83.6%, 26.3%, and 63.2% of these having a ΔAIC >2, for *GFI1B*, *MYB*, and *NFE2* respectively. A similar trend holds even when limiting the models to be fitted to those data points that correspond to a hypothetical one copy loss or gain of the *cis* gene (*Figure 3—figure supplement 6B*), where the median ΔAIC of responsive genes are 7.05, 0.05, and 3.6 for *GFI1B*, *MYB*, and *NFE2 trans* responses. Overall, this shows that *trans* responses to TF dosage are dominated by nonlinear behaviours even when the TF dosage changes are not extreme but within biologically plausible ranges.

## Gene and transcriptional network properties of dosage response

Utilising a model that effectively captures the variance in our data provided the ability to predict unmeasured TF dosage points and facilitated a direct comparison of *trans* effects across different cis genes. Employing the sigmoid model (and loess for those with non-monotonic responses), we estimated the continuous expression of *trans* genes on a uniform fold-change scale across the spectrum of *GFI1B*, *MYB*, and *NFE2* expression changes (*Figure 4A*). This estimation was carried out within the empirically observed range of all three *cis* genes, spanning from log2(FC) –1.83–0.51. Subsequent hierarchical clustering of *trans* gene responses revealed six major clusters of distinct response patterns. For the majority of *trans* genes, the response to *GFI1B* and *MYB* was opposite, with only two small clusters displaying exceptions. Notably, *GFI1B* generally induced the most substantial response, while *NFE2* triggered the smallest range of *trans* gene response.

Next, we collected diverse annotations for the *trans* genes to explore the connections between their regulatory properties, disease associations, and selective constraints concerning their response to TF dosage (*Figure 4B and C*). To quantify these relationships, we assessed significant differences in belonging to these qualitative annotations using the Wilcoxon rank test (*Figure 4D*) and correlated parameters from the sigmoid model with quantitative gene metrics (*Figure 4E*). We hypothesized that genes with annotated selective constraint, numerous regulatory elements, and central positions in regulatory networks would exhibit greater robustness to TF changes. Indeed, housekeeping genes demonstrated a considerably smaller dosage response range (*Figure 4F*). Moreover, genes classified as unresponsive were enriched in the housekeeping category (odds ratio = 2.14, Fisher test *p*-value = 0.024). The link between selective constraint and response properties is most apparent in the *MYB* trans network. Specifically, the probability of haploinsufficiency (pHaplo) shows a significant negative correlation with the dynamic range of transcriptional responses (*Figure 4G*): genes under stronger constraint (higher pHaplo) display smaller dynamic ranges, indicating that dosage-sensitive genes are more tightly buffered against changes in *MYB* levels. This pattern was not reproduced in the other trans networks (*Figure 4E*).

The relationship between the response of *trans* genes and intrinsic gene properties differed between *GFI1B*, *MYB*, and *NFE2 trans* network responses. We also performed a similar analysis comparing the sigmoid parameters to network properties using the approach outlined by *Minaeva et al., 2025* and obtained inconsistent results between TF regulons (*Figure 4—figure supplement 1A and B*). This suggests that the link between commonly annotated gene properties and the gene responses are complex and highly context-specific, as in our data from a single cell line, they differed between the upstream regulators that were manipulated. Thus, much more data is needed before transcriptional responses can be predicted from gene properties, and conversely, to understand the cellular mechanisms that lead to the annotated gene properties.

## Nonlinear dosage responses in complex traits and disease

Moving beyond the characterization of mechanisms of transcriptome regulation, a key question is how gradual dosage variation links to downstream cellular phenotypes, and whether these responses exhibit analogous nonlinear patterns. To address this question, we correlated our findings with the expression profiles of various cell types in order to study the myeloid differentiation process, a phenotype well-characterized for our K562 model that has been used as a reliable system for investigating erythroid differentiation within myeloid lineages (*Wang et al., 2013*) and blood tumours (*Salvadores et al., 2020*). Specifically, leveraging single-cell expression data for bone marrow cell types from the Human Cell Atlas and Human Biomolecular Atlas Project (*Granja et al., 2019*), we filtered the expression data to the targeted genes in our study. After aggregating data across donors and normalising

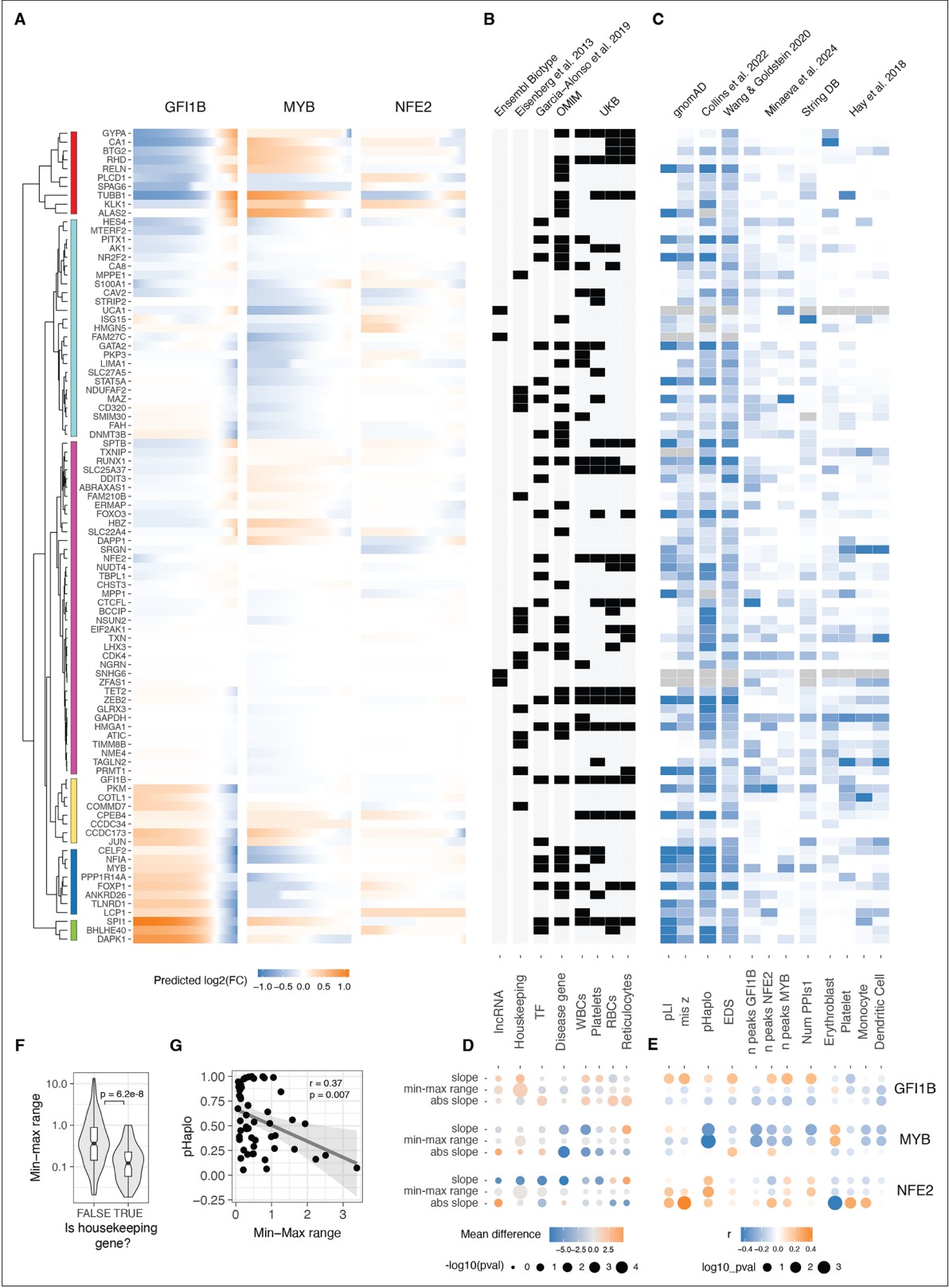

**Figure 4.** Relationship between gene and dosage response properties. (**A**) Predicted changes (using sigmoid or loess fits for monotonic and non-monotonic responses, respectively) in relative expression of all trans genes in response to changes of the GFI1B, MYB, and NFE2 expression. Trans genes (rows) were hierarchically clustered based on their expression fold change linked to alterations of all transcription factors (TFs) dosage. A dendrogram of the resulting clustering shown in the left. (**B**) Heatmap showing the qualitative properties of each trans gene. The x-axis indicates specific

*Figure 4 continued on next page*

*Figure 4 continued*

gene features. The top labels specify the source of the data, while the bottom labels describe the corresponding gene properties. WBCs, platelets, RBCs, and reticulocytes refer to genome-wide association studies (GWAS) of white blood cells, platelets, red blood cells, and reticulocytes, respectively. (**C**) Heatmap indicating the z-scaled quantitative gene features of each transgene. The x-axis indicates specific gene features. The top labels specify the source of the data, while the bottom labels describe the corresponding gene properties. Erythroblast, platelets, monocytes, and dendritic cells refer to cell types from *Hay et al., 2018*. Gray cells indicate missing data. (**D**) The difference in the average value of the sigmoid parameter indicated in the right between the genes qualified into the no/yes category of the gene properties indicated in B. (**E**) Pearson correlation coefficient of the quantitative trans gene features (shown in C) with the sigmoid parameter value for each trans gene in the response to the modulation of dosage of the TF indicated on the left. The size of the points are inversely related to the significance of correlation, and colour indicates the direction of correlation. (**F**) Differences in the range of expression response for Housekeeping vs. non-Housekeeping trans genes with changes of dosage of MYB, GFI1B, and NFE2. (**G**) Negative correlation between haploinsufficiency score (pHaplo) and the range of the response of trans genes to the modulation of MYB.

The online version of this article includes the following figure supplement(s) for figure 4:

**Figure supplement 1.** Relationship of gene properties and transcription factor (TF)-target network properties with TF dosage responses.

expression across cell types (*Figure 5—figure supplement 1A*), we compared the expression patterns resulting from each unique transcription factor dosage modulation in relation to each unique cell type expression state. The ensuing correlation can then be construed as a 'phenotype,' signifying the similarity between the transcriptional state induced by the TF increase or decrease and the transcriptional state of a specific blood lineage cell type.

Such analyses recapitulate known biology, with *GFI1B* upregulation (*Möröy et al., 2015*) and *MYB* downregulation (*Jayapal et al., 2010*) being crucial factors promoting erythrocyte maturation (*Figure 5A*). The downregulation of *NFE2*, instead, was negatively related to platelet differentiation. Analysing the correlations as inferred phenotypes suggests potential nonlinear relationships (*Figure 5—figure supplement 1B*), but these trends should be considered hypotheses that require experimental validation. In summary, this points to cellular phenotypes resulting from gradual TF dosage modulation.

Many of the analyzed *trans* genes are associated with physiological traits and diseases (*Figure 4*). Understanding the nonlinear trends in the expression of these genes is of particular interest, as it helps comprehend how genes with physiological impacts may be buffered against upstream regulatory changes, and how their dosage changes as a response to upstream regulators contrasts with genetic variants that contribute to diseases and traits. Additionally, knowing the underlying dosage-to-phenotype curve of a gene can be crucial if this is considered a biomarker for identifying or treating disease. To investigate this, we analyzed whether OMIM genes for rare diseases and Mendelian traits or GWAS genes for different blood cell traits (*Figure 5B*) that are part of the *trans* networks of genes affected by *GFI1B*, *MYB*, or *NFE2* perturbation are enriched for nonlinear dosage responses. As seen in *Figure 4*, the *trans* response properties of each gene are highly specific to the regulators and thus analyzed in parallel for each *cis* gene network. An enrichment for nonlinear responses was observed for *MYB trans* network genes associated with disease and for blood traits related to white blood cells and reticulocytes. These enrichments are particularly interesting given that most *trans* genes that were sensitive to *MYB* dosage modulation did not respond with a nonlinear trend (*Figure 3—figure supplement 6A*).

Despite nonlinear responses not being significantly enriched among disease genes across all *trans* networks, the responses of the same *trans* gene can show very different dosage responses depending on the upstream regulator being tuned. In *Figure 5C*, we highlight several disease-associated genes (linked to one or more disease phenotypes *Amberger et al., 2015*). *FOXP1*, a haploinsufficient and potentially triplosensitive transcription factor implicated in intellectual disability, exhibited a strong and dose-dependent response, particularly to varying levels of *GFI1B*. A similar pattern was observed for *NFIA*, another haploinsufficient gene involved in developmental disorders. However, it is difficult to interpret their expression response in K562 cells when their most apparent phenotypic effects likely derive from other cell types. *RHB* is the Rhesus blood type gene, where a common deletion of the gene causes the Rh blood type in homozygous individuals, with a strong nonlinear response to *GFI1B* levels. A particularly interesting gene is *TUBB1*, part of β-tubulin, that causes autosomal dominant macrothrombocytopenia or abnormally large platelets. Here, K562 cells are a reasonable model system, being closely related precursors to megakaryocytes that produce platelets. Interestingly, *GFI1B* loss also causes a macrothrombocytopenia phenotype in mice (*Beauchemin et al., 2017*),

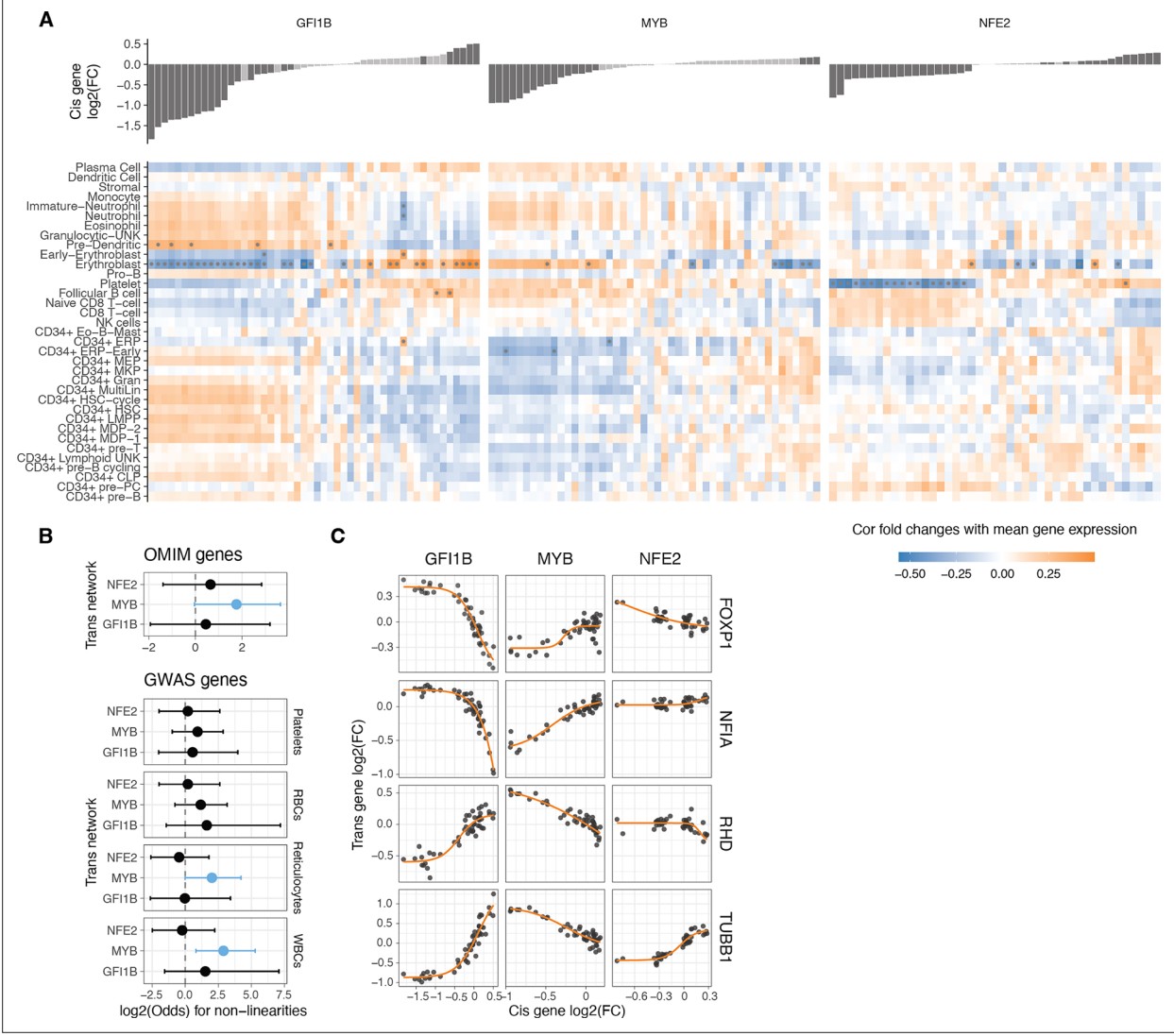

**Figure 5.** Non-linearities in transcription factor (TF) dosage responses of complex traits and disease genes. (**A**) Heatmap illustrating the correlation between the mean expression of cell types and the changes in expression linked to individual TF dosage perturbations. The bar plot on the top panel represents cis gene dosage perturbation. Asterisks (*) denote correlations with 10% FDR. (**B**) Enrichment log(odds) ratio of non-linear TF dosage responses ($\Delta AIC_{linear-sigmoid} > 0$) in disease-related genes (OMIM genes linked to 1 or more diseases, top panel) or in GWAS blood traits-associated genes (closest expressed gene to lead GWAS variant, bottom panel). Log(odds) with Fisher's exact test at FDR <0.05 are highlighted in blue. (**C**) Examples of TF dosage response curves of genes both associated with disease (OMIM) and complex traits (Blood GWAS).

The online version of this article includes the following figure supplement(s) for figure 5:

**Figure supplement 1.** Transcriptional similarity among bone marrow cell types at different transcription factor (TF) dosage levels.

and in our data, *TUBB1* expression decreases quickly as a function of decreased *GFI1B* expression but then plateaus at a level that corresponds to loss of one copy of *TUBB1*. This raises the hypothesis that low *GFI1B* levels may cause macrothrombocytopenia at least partially via reducing *TUBB1* expression.

## Discussion

In this paper, we have investigated how gradual dosage modulation of transcription factors contributes to dosage-sensitive transcriptional regulation and investigated its potential phenotypic consequences. First, we set up an easily scalable and generalizable CRISPRi/CRISPRa approach with tiling sgRNAs for gradual titration of gene expression, with reagents that can be designed with data only of the TSS and easily ordered at scale. Alternative approaches that rely, e.g., on targeting

CREs that are often unknown, dramatic overexpression, or laborious setting up of constructs for each gene are less practical for large-scale analyses. Our approach appears best suited for expression modulation in the biologically reasonable range, and other methods would be needed for dramatic overexpression or complete silencing of the target genes. Our inability to substantially increase *MYB* expression indicates the need for further work and larger *cis* gene sets to fully understand how widespread this is and to what extent this depends on *cis*-regulatory properties versus feedback and buffering mechanisms. Nevertheless, we believe that the approach proposed here is a useful complement to the diversifying set of tools for dosage modulation for different purposes (*Jost et al., 2020*; *Hawkins et al., 2020*; *Noviello et al., 2023*; *Chiarella et al., 2020*; *Liu et al., 2024*).

In this work, we made use of targeted transcriptome sequencing to avoid complications from the sparsity of single-cell data. While highly accurate targeted readout of the *cis* gene expression linked to each sgRNA is a core component of our approach, analysing *trans* responses could also be achieved by standard single-cell sequencing of the full transcriptome, possibly in combination with a targeted readout of transcripts of particular interest. In this study, the targeted genes were selected based on prior data of responding to *GFI1B*, *NFE2,* or *MYB* regulation and thus do not represent an unbiased or random sample of genes. An interesting future extension would be the addition of single-cell protein quantification to confirm that the detected mRNA levels correspond to protein levels, but this remains technically challenging.

Our results show that nonlinear responses to gradual up- and down-regulation of TF dosage are widespread and can be detected even without extreme overexpression or full knockout of the TF. The patterns of transcriptional responses are highly context-specific and vary between upstream regulators. Further work with larger sets of *cis* and *trans* genes, as well as direct quantification of cellular readouts, will be needed to fully characterise the patterns and mechanisms of downstream impacts on gene dosage. However, our findings indicate important directions for future research. First, the widespread nonlinearity suggests that inferring gene function from classical molecular biology approaches—such as drastic knockouts or knockdowns—may be limited, as these perturbations can produce effects that are both quantitatively and qualitatively different from the more modest changes that occur naturally. This may be particularly relevant for essential and highly dosage-sensitive genes, where applying our gradual dosage modulation framework can provide opportunities for functional characterization at perturbation levels that do not kill the cells. Second, we show that the effects of up- and downregulation are qualitatively and quantitatively different, which calls for increased attention to analysing both directions of effect, which also occur in natural responses to genetic variants and environmental stimuli.

Gene dosage sensitivity has typically been studied by human genetics and genomics methods (*Collins et al., 2020*; *Mohammadi et al., 2019*; *Dong et al., 2023*). The experimental approach pursued in this study and the computational approaches are fundamentally different and complement each other: while human genetics is powerful for capturing the functional importance of physiological phenotypes via patterns of population variation and selective constraint, experimental approaches provide more granularity and insights into cellular mechanisms. Furthermore, while the convergence of disease effects of common and rare variants affecting the same gene is a well-known phenomenon (*Backman et al., 2021*; *Freund et al., 2018*), the sparsity of variants makes it difficult to properly model allelic series as a continuous dosage-to-phenotype function for individual genes. Experimental approaches can provide a powerful complement to this. Altogether, we envision that combining these perspectives into true systems genetics approaches will be a powerful way to understand how gene dosage variation contributes to human phenotypes from molecular to cellular and eventually physiological levels.

# Materials and methods

## Key resources table

| Reagent type (species) or resource | Designation | Source or reference | Identifiers | Additional information |
|---|---|---|---|---|
| Recombinant DNA reagent | pCC_05: Lentiviral puromycin CRISPRa dCas9-VPR system | Addgene | RRID:Addgene_139090 | Used as PCR template for dCas9-VPR cassette (*Legut et al., 2020*). |
| Recombinant DNA reagent | pGC02: Lentiviral blasticidin CRISPRi KRAB-dCas9-MeCP2 system | other | RRID:Addgene:_170068 | Sourced from Sanjana Laboratory (*Morris et al., 2023*). Backbone for pJDE003 construction; digested with XbaI-FD and BamHI-FD. |
| Recombinant DNA reagent | pJDE003: Lentiviral blasticidin CRISPRa dCas9-VPR system | this study | NA | Constructed by replacing KRAB-dCas9-MeCP2 cassette in pGC02 with dCas9-VPR PCR product from pCC_05; Gibson assembled (2:1 insert:vector). |
| Recombinant DNA reagent | pGC03: Lentiviral puromycin sgRNA library cloning vector | Addgene | RRID:Addgene:_170069 | Used for cloning 96-sgRNA library (BsmBI digestion; NEBuilder HiFi assembly). |
| Recombinant DNA reagent | pMD2.G: Lentiviral envelope plasmid | Addgene | RRID:Addgene:_12259 | Envelope plasmid for lentiviral production. |
| Recombinant DNA reagent | psPAX2: Lentiviral packaging plasmid | Addgene | RRID:Addgene:_12260 | Packaging plasmid for lentiviral production. |
| strain, strain background (*Escherichia coli*) | NEB 5-alpha competent cells | New England Biolabs | NEB:C2987H | Used for plasmid transformations (pJDE003 assemblies). |
| Strain, strain background (*E. coli*) | One Shot Stbl3 chemically competent cells | Thermo Fisher Scientific | ThermoFisher:C737303 | Used for cloning/propagating lentiviral vectors. |
| Strain, strain background (*E. coli*) | Endura electrocompetent cells | Lucigen | Lucigen:60242–2 | Used for sgRNA library transformation by electroporation;>2.5e5 transformants obtained. |
| Cell line (Human) | HEK293FT | Thermo Fisher Scientific | ThermoFisher:R70007; RRID:CVCL_6911 | Maintained at 37 °C, 5% CO2 in DMEM high glucose (Cytiva SH30022.01)+10% Serum Plus II (Sigma 14,009 C). |
| Cell line (Human) | K562 | ATCC | ATCC:CCL-243; RRID:CVCL_0004 | Maintained at 37 °C, 5% CO2 in IMDM, GlutaMAX (ThermoFisher:31980097)+10% Serum Plus II (Sigma 14,009 C). |
| Antibody | Purified anti-CRISPR (CAS9) antibody (clone 7 A9) | BioLegend | BioLegend:844302; RRID:AB_2749904 | Primary antibody for western blot of dCas9 (conditions not specified in excerpt). |
| Antibody | GAPDH (14 C10) Rabbit monoclonal antibody | Cell Signaling Technology | CST:2118 S; RRID:AB_561053 | Primary antibody for loading control western blot (conditions not specified in excerpt). |
| Antibody | IRDye 800CW goat anti-mouse IgG (H+L) | LI-COR | LI-COR:925–32212 | Secondary antibody for CAS9 western blot (conditions not specified in excerpt). |
| Antibody | IRDye 680RD goat anti-rabbit IgG (H+L) | LI-COR | LI-COR:925–68073 | Secondary antibody for GAPDH western blot (conditions not specified in excerpt). |
| Antibody | FITC anti-human CD4 antibody (clone RPA-T4) | BioLegend | BioLegend:300505; RRID:AB_314073 | Used for FACS validation of CRISPRa activation (day 4 and day 10/11 post-transduction). |
| antibody | APC anti-human CD19 antibody (clone HIB19) | BioLegend | BioLegend:302211; RRID:AB_314241 | Used for FACS validation of CRISPRa activation. |
| Antibody | PE anti-human CD45 antibody (clone 2D1) | BioLegend | BioLegend:368509; RRID:AB_2566369 | Used for FACS validation of CRISPRa activation. |
| Commercial assay or kit | Q5 High-Fidelity 2 X Master Mix | New England Biolabs | NEB:M0492L | PCR amplification of dCas9-VPR cassette. |
| Commercial assay or kit | Gibson Assembly Master Mix | New England Biolabs | NEB:E2611S | Used for Gibson assembly (2:1 insert:vector). |

*Continued on next page*

*Continued*

| Reagent type (species) or resource | Designation | Source or reference | Identifiers | Additional information |
|---|---|---|---|---|
| Commercial assay or kit | NEBuilder HiFi DNA Assembly kit | New England Biolabs | NEB:NEBuilder-HiFi | Used for cloning pooled sgRNA library into BsmBI-digested pGC03 (10 reactions). |
| Commercial assay or kit | Plasmid Maxiprep Kit | QIAGEN | QIAGEN:12362 | Used for plasmid DNA preparation for virus production. |
| Commercial assay or kit | Maxi Fast-Ion Plasmid Kit, Endotoxin Free | IBI Scientific | IBI:IB47123 | Used for sgRNA library plasmid maxiprep. |
| Commercial assay or kit | Steriflip-HV 0.45 µm filter | Millipore | Millipore:SE1M003M00 | Filtration of harvested lentiviral supernatant. |
| Commercial assay or kit | Lentivirus Precipitation Solution | Alstem | Alstem:VC100 | Used for lentiviral concentration (10 X or 2 X as described). |
| Commercial assay or kit | 10 x Chromium Next GEM Single Cell 5' Reagent Kit v2 (single indexing) | 10 x Genomics | 10 x:PN-1000265; 10 x:PN-1000190 | Used for 5' single-cell library prep (two lanes; ECCITE-seq modifications). |
| Commercial assay or kit | 10 x Targeted Gene Expression protocol | 10 x Genomics | 10 x:PN-1000248 | Custom probe library used for targeted enrichment of genes of interest. |
| Commercial assay or kit | Illumina NextSeq 500/550 Mid Output v2.5 kit (150 cycles) | Illumina | Illumina:NextSeq-MidOutput-v2.5–150 | Sequencing of targeted gene expression, HTO and GDO libraries. |
| Commercial assay or kit | Illumina MiSeq Reagent Kit v3 (150 cycles) | Illumina | Illumina:MiSeq-v3-150 | Sequencing of dCas9 targeted enrichment and additional HTO libraries. |
| Commercial assay or kit | xGen Custom Hybridization Capture Panel (biotinylated oligos) | IDT | IDT:xGen-Custom-Panel | Custom targeted gene expression panel (final 4,405 probes;~15% discarded during design). |
| Commercial assay or kit | LookOut Mycoplasma PCR Detection Kit | Sigma-Aldrich | Sigma:MP0035 | Routine mycoplasma testing (frequency not specified). |
| Peptide, recombinant protein | XbaI FastDigest (XbaI-FD) | Thermo Fisher Scientific | ThermoFisher:FD0685 | Restriction digest of pGC02. |
| Peptide, recombinant protein | BamHI FastDigest (BamHI-FD) | Thermo Fisher Scientific | ThermoFisher:FD0054 | Restriction digest of pGC02. |
| Peptide, recombinant protein | FastAP Thermosensitive Alkaline Phosphatase | Thermo Fisher Scientific | ThermoFisher:EF0651 | Vector dephosphorylation after restriction digest. |
| Peptide, recombinant protein | DpnI | Thermo Fisher Scientific | ThermoFisher:FD1704 | Digest PCR template plasmid (15 min) prior to Gibson assembly. |
| Other | DMEM high glucose with L-glutamine; without sodium pyruvate | Cytiva (HyClone) | Cytiva:SH30022.01 | Used for HEK293FT culture and lentivirus resuspension media. |
| Other | IMDM, GlutaMAX | Thermo Fisher Scientific | ThermoFisher:31980097 | Used for K562 culture. |
| Other | Serum Plus II medium supplement | Sigma-Aldrich | Sigma:14,009 C | Used at 10% supplementation for HEK293FT and K562 culture. |
| Chemical compound, drug | Polyethylenimine (PEI) linear MW 25,000 | Polysciences | Polysciences:23966 | Used for HEK293FT transfection for lentivirus production. |
| Chemical compound, drug | Blasticidin | A.G. Scientific | A.G.Scientific:B-1247 | Used at 10 µg/mL for 16 days to select dCas9-VPR K562 clones; also 5 µg/mL during sgRNA library culture as described. |

*Continued on next page*

*Continued*

| Reagent type (species) or resource | Designation | Source or reference | Identifiers | Additional information |
|---|---|---|---|---|
| Chemical compound, drug | Puromycin | InvivoGen | InvivoGen:ant-pr-1 | Used at 2 µg/mL for sgRNA integration selection. |
| Chemical compound, drug | GlycoBlue | Thermo Fisher Scientific | ThermoFisher:AM9515 | Used for DNA precipitation of pooled sgRNA library assemblies. |
| Chemical compound, drug | Isopropanol | other | NA | Used for DNA precipitation of pooled sgRNA library assemblies. |
| Chemical compound, drug | NaCl | other | NA | Used at 50 mM during DNA precipitation of pooled sgRNA library assemblies. |
| Chemical compound, drug | Ethanol 70% | other | NA | Used for washes during DNA precipitation cleanup. |
| Sequence-based reagent | PCR primers oJDE005 and oJDE006 | this study | NA | Used to amplify dCas9-VPR cassette from pCC_05; primer sequences not provided in excerpt. |
| Sequence-based reagent | 96-sgRNA library (ssDNA oligos, 60 bp) for gene dosage library | IDT | IDT:ssDNA-oligos-plate | 96 guides pooled equimolarly to 0.2 µM; cloned into pGC03; guide sequences not provided in excerpt. |
| Software, algorithm | FastQC | Babraham Bioinformatics | RRID:SCR_014583 | Used for QC/demultiplexing of FASTQs (version not specified). |
| Software, algorithm | Cell Ranger (cellranger count) | 10 x Genomics | RRID:SCR_017344 | Used for gene expression (with targeted-panel) and guide capture analysis (Gaussian mixture model calling). |
| Software, algorithm | Seurat | *Hao et al., 2021* | RRID:SCR_016341 | Used for normalization, scaling, and UMAP; Seurat v4.3 used for NormalizeData and downstream analyses. |
| Software, algorithm | Salmon/Alevin | *Srivastava et al., 2020* | RRID:SCR_017036 | Used for HTO quantification (version not specified). |
| Software, algorithm | Sceptre | *Barry et al., 2021* | NA | Used to validate calibration of control cells (*Figure 1e*). |
| Software, algorithm | R (stats: lm, loess, AIC; drc: drm(fct =L.4())) | R Foundation; drc package | RRID:SCR_001905 | Used for model fitting (linear, LOESS, 4-parameter sigmoid) and AIC calculation. |

## CRISPRa vector construction

To construct the vector harbouring the CRISPRa system (pJDE003), the CRISPRi (KRAB-dCas9-MeCP2) gene fusion of pGC02 was replaced with dCas9-VPR cassette, which was PCR amplified (Q5 High-Fidelity 2X Master Mix, NEB M0492L) from the plasmid pCC_05 (*Morris et al., 2023*) with primers oJDE005 and oJDE006 following instructions from manufacturer. pGC02 was digested with XbaI-FD and BamHI-FD (Thermo Fisher FD0685 and FD0054) and sequentially dephosphorylated with FastAP (Thermo Fisher EF0651) following the manufacturer's recommendations. The digested pGC02 vector and the PCR insert with the CRISPRa system (previously treated with a 15 min DpnI enzyme incubation, Thermo Fisher FD1704) were assembled by Gibson assembly using a 2:1 insert:vector ratio with Gibson Assembly Master Mix (NEB E2611S). Assemblies were transformed into NEB 5-alpha *E. coli* competent cells and single colonies were picked and sequence validated by Sanger sequencing. Frozen stock of the correct construct cells were regrown for plasmid Maxiprep extraction (QIAGEN 12362) for subsequent virus production.

## CRISPRa K562 cell line construction and functional validation

Lentivirus was produced by polyethylenimine linear MW 25000 (Polysciences 23966) transfection of HEK293FT cells with the transfer plasmid containing a Cas9-VPR effector, packaging plasmid psPAX2 (Addgene 12260) and envelope plasmid pMD2.G (Addgene 12259). After 72 hr post-transfection, cell media containing lentiviral particles was harvested and filtered through 0.45 µm filter Steriflip-HV (Millipore SE1M003M00). One volume of Lentivirus Precipitation Solution (Alstem VC100) was added

to the collected lentivirus, mixed and stored overnight at 4°C. The mix was centrifuged for 30 min at 1,500g, and the pellet of lentiviral particles were resuspended in 1/10th of the original volume of DMEM media. Lentivirus vials were frozen at -80°C and later thawed for transduction.

To construct the monoclonal K562 cell line with the CRISPRa system, the dCas9-VPR lentivirus was transduced into one million K562 cells using 100 µl of 10X concentrated lentivirus in a total volume of 1 ml (high MOI). After 24 hr, the media was replaced with fresh IMDM, and 48 hr after transduction, blasticidin (A.G. Scientific B-1247) was added to a final concentration of 10 µg/µl for 16 days. Monoclonal cell lines were sorted by FACS (Sony Cell Sorter SH800) into a 96-well plate. The presence of dCAS9 protein in several growing clones was confirmed by western blot (Primary antibody: Purified anti-CRISPR CAS9 antibody; BioLegend 844302, Secondary antibody: LI-COR 925-32212) and protein levels were normalized to GAPDH (Primary antibody: GAPDH (14C10) Rabbit; Cell Signalling Technology 2118S. Secondary antibody: LI-COR 925-68073).

To select the final monoclonal CRISPRa cell line, the three clones with the highest protein expression in the western blot were subjected to functional validation to test for activation activity. Lentiviral guides designed from *Legut et al., 2020* targeting CD4 (Anti-CD4 Mouse Monoclonal Antibody (FITC), BioLegend, 300505), which is lowly expressed in K562, CD19 (Anti-CD19 Mouse Monoclonal Antibody (APC), BioLegend, 302211) with null expression, and CD45 (PE anti-human CD45, BioLegend, 368509) with intermediate expression, were independently transduced into all three monoclonals, and after puromycin selection, the expression of this markers was screened by FACS at day 4 and at day 10 or 11 after transduction. The clone with the strongest and most consistent activity was selected. K562 cells were obtained as fresh stocks from ATCC for this work and not authenticated further. Cells were regularly passaged and tested for the presence of mycoplasma contamination with a Mycoplasma PCR Lookout assay.

## Gene selection for targeted sequencing and design of probe custom panel

The four selected dosage genes were GFI1B, NFE2, MYB, and TET2. GFI1B, and NFE2 were chosen due to their reported trans effects following the inhibition of their cis-CREs (*Morris et al., 2023*). MYB was selected for being downstream of the GFI1B network, and TET2 was selected as an unrelated gene to those transcriptional networks. Both MYB and TET2 qualified as oncogene and tumour suppressor functions in K562 (https://depmap.org/portal/), making them ideal choices to determine the impact of growth effects on the experiment.

The additional 88 genes captured by targeted sequencing were selected based on the significant trans effects of GFI1B and NFE2 inhibition from *Morris et al., 2023* including two control genes, GAPDH and LHX3 genes, one highly constantly expressed 'housekeeping gene' and the other with no reported expression in K562 cell lines, respectively. The remaining 86 genes were selected based on Morris et al. to include (1) 29 genes that overlapped between the NFE2 and GFI1B network, (2) 47 trans genes for GFI1B, 10 trans genes for NFE2. The number of trans genes selected from each unique network was proportionally chosen, given the size of each trans network, and oversampling TFs and TF targets as defined in *Garcia-Alonso et al., 2019*, as well as maintaining the proportional co-expression cluster structure as defined in *Morris et al., 2023*; *Figure 1—figure supplement 1A and B*. Additional filters in the selection included a minimum expression of 0.1 mean UMI/cell and the lack of alternative 5' splice isoforms and a unique Ensembl ID.

The 10X Probe Full Custom Panel design tool was used to design the targeted gene expression probe library. A total of 687 probes (~15%) were discarded because they covered transcript regions with a median coverage per base of <3 reads/bp (for medium and highly expressed genes) or <1 reads/bp (for lowly expressed genes). All probes for LHX3 (0 median reads/bp) were retained. In addition, 93 probes covering the entire transcript sequence of the dCas9-VPR and KRAB-dCas9-MeCP2 transcript were included, resulting in a final total of 4405 probes. The xGen Custom Hybridization Capture Panel of biotinylated oligos was ordered and synthesized at IDT.

## Gene dosage sgRNA library design and cloning

The sgRNA library contained a total of 96 guides (51 tiling, 8 TSS, 20 attenuated, 12 enhancer, and 5 non-targeting controls). All guides were designed to not contain the U6 terminator sequence, repeats of five or more consecutive G, C, or As, as well as not falling in the genomic region where K562 cell

line has alternative alleles compared to the human genome reference (Hg38). All guides were scored with FlashFry *McKenna and Shendure, 2018* to obtain off-target and on-target activity scores that allowed the selection of the best scoring guides. Tiling guides were designed to target different regions of the promoter, TSS and beginning of the gene body of each dosage gene, spanning a total average distance of 1400 bp (TSS in the centre), each being on average distant from one another of 110 bp. The sequences of the two TSS guides were obtained from *Replogle et al., 2022*. The sgRNAs targeting enhancers were picked from previously reported work that showed a CRISPR-based evidence of enhancer activity (GFI1B *Morris et al., 2023*), (NFE2 *Xie et al., 2019*), (MYB *Li et al., 2021*). The five attenuated guides for each gene were manually designed following the rules described in *Jost et al., 2020* to span a range of activities, including a single point mutation on the best scoring guide that targeted the TSS.

Overhangs with homology regions to the pGC03 plasmid (18 bp downstream and 22 bp upstream) were added to the sgRNA sequence to be able to directly clone the ssDNA oligos into the plasmid. The 96 sgRNAs were ordered in IDT as single-stranded DNA oligos (total 60 bp) in a 96-well plate to 100 pmol scale. The oligos were pooled at equimolar concentration and diluted to a final concentration of 0.2 µM. The library was cloned into the BsmBI-digested plasmid pGC03 using 10 reactions of the NEBuilder HiFi DNA Assembly kit following the manufacturer's instructions. All the reactions were pooled and the DNA precipitated using isopropanol, GlycoBlue (Thermo Scientific AM9515) and 50 mM NaCl for 15 min at RT. Following two washes with ethanol 70%, the assembly was resuspended with 15 µl of 0.2X TE.

To transform the library into *E. coli*, 1 µl of the assembly was mixed with 25 µL of Endura cells under manufacturer's electroporation conditions, then plated onto 245 × 245 mm square LB 100 µg/ml Carbenicillin plates. The plates were grown ON, and >2.5e5 transformants were obtained, ensuring the complexity of the library was maintained at >1000 cells per unique sgRNA. All colonies were collected and subjected to maxiprep using the Maxi Fast-Ion Plasmid Kit, Endotoxin Free kit (IBI Scientific IB47123). The representation of the library was assessed through MiSeq shallow sequencing (Illumina).

## sgRNA lentiviral library production and cell culture assay

The lentiviral library was produced by transfecting ~80 million HEK293FT cells with a transfer plasmid containing the 96 sgRNA library, along with the packaging plasmid psPAX2 and envelope plasmid pMD2.G, using polyethylenimine linear MW 25000. The supernatant media was replaced with fresh D10 10% BSA 6 hr after transfection, and the virus was collected and filtered through 0.45 µm filters after 48 hr. The lentiviral library was then concentrated 2X using the Precipitation Lentiviral Solution, aliquoted, and stored at -80°C for subsequent transduction.

Both CRISPRi and CRISPRa K562 cell lines were independently transduced with different titers of the lentiviral sgRNA library at a low MOI (one sgRNA per cell). Twenty hours post-transduction, the cell media was replaced with fresh IMDM 10% Serum Plus II Blasticidin 5 µg/mL, and 4 hr later, Puromycin (Invivogen ant-pr-1) was added at a final concentration of 2 µg/mL to select for cells with sgRNA integration. The transduction batch with an infection rate of ~10% was selected, and cells were sorted to near purity using FACS to remove dead cells. Cells were maintained at >90% survival and a maximum confluency of 700,000 cells/mL. On day 8 post-transduction, the cells were collected and prepared for cell hashing.

## Multimodal single-cell experiment and targeted sequencing

Cell hashing was performed as previously described using four hashtag-derived oligonucleotides (HTOs) in a hyperconjugation protocol (*Stoeckius et al., 2017*). Each transduced cell line was split into four batches of 500,000 cells, resulting in a total of 8 different hashes. After incubation and washes, all 8 hashed batches were pooled together and run in two reaction lanes of the 10X Chromium Next GEM Single Cell 5' Reagent Kit v2 (single indexing, PN-1000265, and PN-1000190). The manufacturer's protocol was followed with modifications stipulated in the ECCITE-seq protocol (*Mimitou et al., 2019*). For each GEM reaction, 42,000 cells from the hash pool were used to obtain approximately 21,000 total cells, including 'multiplets' (multiple cells per droplet counts). Gene expression (cDNA), hashtags (HTOs), and guide RNA (Guide-derived oligos, GDOs) libraries were constructed following the 10x Genomics and ECCITE-seq protocols (https://cite-seq.com/eccite-seq/) with minimal

modifications. Specifically, the antibody pool protein tag library steps were ignored, and a custom-designed probe library was used to enrich the cDNA for the genes of interest in the 10X Targeted Gene Expression protocol (10X PN-1000248). The resulting libraries were sequenced using an Illumina Nextseq 500/550 Mid-Output v2.5 Kit (150 cycles). The targeted enrichment of the dCas9 transcripts was performed separately using an independent probe library and was sequenced together with additional HTO libraries using the Illumina Miseq Reagent Kit v3 (150 cycles).

## Computational and statistical analyses

### From fastqs to QCed and demultiplexed UMI normalized matrices

FastQC was used to demultiplex the different samples of the three different modalities from the different 10X chip lanes, which each was processed independently. For the cDNA modality, the UMI count matrix was obtained using Cellranger *count*, including the *targeted-panel* argument to get the additional filtered matrices and summary statistics. Cells with less than 500 UMIs per cell or less than 50 genes with at least 1 UMI per cell were discarded. The top 1% of cells containing the highest UMI content were also discarded. The expression of all genes across 10X lanes were extremely reproducible (Pearson r = 0.999), showing a ~fivefold UMI count increase in contrast to the non-targeted transcriptome (*Figure 1—figure supplement 1C*).

For the GDO modality (sgRNAs), Cellranger count was also used using the CRISPR Guide Capture Analysis mode, which uses a Gaussian mixture model to call sgRNA per cell. The cells containing more than one sgRNA were discarded.

To classify each cell into its corresponding CRISPR system of origin (CRISPRi or CRISPRa), both the HTO modality (protein hashes) and the expression of the dCas9 targeted transcript was used. Protein hashes were called using Alevin salmon (*Srivastava et al., 2020*), and the resulting HTO UMI matrix was mixed in with the cDNA matrix containing the expression of the CRISPRi and CRISPRa genes. This matrix was normalized and scaled using Seurat v4 (*Hao et al., 2021*) and used to generate a UMAP based on the expression of protein hashes and the dCas9 transcripts expression. Clusters were identified and manually assigned to an HTO category given the expression pattern of each cluster. Finally, the 5% of cells classified as CRISPRi that had the lowest expression of CRISPRi transcript were discarded, as well as those 5% of CRISPRa classified cells that had the highest CRISPRi transcript expression. In total, 20,001 (10,647 CRISPRi and 9354 CRISPRa) cells passed all filters and were used for subsequent analyses.

Once each single cell was classified into a unique sgRNA perturbation and to a cell line of origin, the cDNA UMI matrices of the two 10X lanes were merged and afterwards normalized using the log1p normalization method of Seurat's NormalizeData (Seurat version 4.3). On average, each unique sgRNA perturbation was measured in 81 and 86 cells for the CRISPRa and CRISPRi, respectively (*Figure 1—figure supplement 1D*).

### Expression fold-change calculation and non-target sgRNA filtering

As estimates of changes in expression, we used a pseudo-bulk differential analyses approach. To get rid of the batch effects deriving from each cell line (CRISPRa vs. CRISPRa) (*Figure 3—figure supplement 1A*), for each unique perturbation, we calculated log2 fold-change of the expression of a gene against the expression of that gene in the population of cells harbouring the NTC sgRNAs of that particular cell line.

Before running the differential analyses on all targeted genes, across all unique CRISPR perturbations, we identified those NTC sgRNAs that had potential unexpected off-target activity and thus could not be used as negative controls. For all possible unique NTC sgRNA pairs, we run the above differential expression analysis on all 92 targeted genes. We discarded NTC sgRNAs that showed more than one DE gene (FDR 0.05) in more than one pairwise comparison, and the differential genes showed consistent patterns of change in expression. For this reason, cells harbouring sgRNA NTC_2 on the CRISPRa modality were discarded, as this particular perturbation showed consistent undesired activation of PPP1R14A and CTCFL genes. Additionally, we ran Sceptre (*Barry et al., 2021*) using the resulting group of control cells to validate that our samples were calibrated correctly (*Figure 1—figure supplement 1E*).

Once those potential outlier NTCs were discarded, the log2FC of each targeted gene in each unique sgRNA and cell line condition was calculated. We used Seurat's FindMarkers() function, which

computes the log fold change as the difference between the average normalized gene expression in each group on the natural log scale:

Logfc = log_e(mean(expression in group 1)) - log_e(mean(expression in group 2))

This is calculated in pseudobulk where cells with the same sgRNA are grouped together and the mean expression is compared to the mean expression of cells harboring NTC guides. To calculate per-gene differential expression p-value between the two cell groups (cells with sgRNA vs cells with NTC), the Wilcoxon Rank-Sum test was used. Adjusted FDR p-values were calculated across all tests to later on call significance on DE genes. The obtained fold changes and FDRs were used for all subsequent analyses.

## Linear, loess, and sigmoidal model fitting

To identify the best predictive model of each cis-gene dosage to trans-gene fold change, we fitted three types of models to the data: linear (using the R *lm* function), a four-parameter sigmoid (using the *drm(fct = L.4())* function from the R dcr package) and a LOESS fit (R *loess* function). To evaluate and compare the goodness of fit of the linear vs. the sigmoid model taking into account overfitting, we calculated the Akaike information criterion (AIC) using the *AIC* function from the R stats package.

To obtain an accurate prediction of each trans gene expression given TF dosage and avoid over-fitting, a 10-fold cross-validation scheme was followed by fitting the sigmoid model individually to each curve. The data was randomly split into 10 groups, where 90% of the data was used for training and the remaining 10% for testing. To obtain the values of each individual sigmoid fit for each dosage and trans gene response, the average and standard deviation of each parameter value was calculated across the 10 trained models.

Those trans genes with a slope significantly different from 0 (FDR adjusted p-value of a z-test across the 10-fold-CV parameter outputs) and with a min-to-max range significantly higher than 0.05 (FDR adjusted p-value of a z-test across the difference between the min and max asymptotes parameters in the 10-fold-CV), were classified as 'responsive' genes. The remaining genes were classified as 'unre-sponsive.' The top 5% trans genes of the GFI1B trans network with the largest $\Delta$RMSE between the LOESS fit and the sigmoid fit ($RMSE_{Sigmoid}$ - $RMSE_{LOESS}$) were classified as non-monotonic and the curve trend manually validated. For the five trans genes classified to have a non-monotonic gene expression response, their predicted expression upon TF dosage change was calculated using the LOESS model instead of the sigmoid one.

## Gene-specific properties

Diverse gene annotations and properties were collected to compare with the different *trans* genes response properties (related to *Figure 4*). Quantitative annotations included the gene biotype (Ensembl) *Martin et al., 2023*, Housekeeping genes (*Eisenberg and Levanon, 2013*), transcription factors (*Garcia-Alonso et al., 2019*), genes associated with at least one disease (OMIM *Amberger et al., 2015*), and genes associated with blood-related complex traits (obtained from *Morris et al., 2023*).

Quantitative features included the probability of being loss-of-function intolerant scores (pLI) (*Lek et al., 2016*) and synonymous and missense Z scores (mis z) (*Lek et al., 2016*; *Samocha et al., 2014*), which were obtained from the GnomAD database. Haploinsufficiency probability scores were obtained from *Collins et al., 2022*. To obtain the number of ChIP-Seq peaks of a *cis* gene within the promoter region of trans-genes (n peak [cis gene]), we utilized the regulon generated by *Minaeva et al., 2025*. This regulon was created by mapping transcription factor peaks to transcription start sites (TSS) of the 50% expressed isoforms for each gene in K562 cells, with subsequent application of a ±1 Kb proximity filter. Mean expression of genes from bone marrow cell types were obtained from *Hay et al., 2018* and averaged across donors. The number of protein-protein interactions of each gene within the entire human proteome (Num PPIs1) was obtained from the STING database (*Szklarczyk et al., 2015*).

To test significant differences between groups of genes (qualitative features), the Wilcoxon rank test was used. For quantitative features, Pearson correlation between parameters from the sigmoid model with quantitative gene metrics was used. Non-responsive and non-monotonic genes in each trans network were excluded.

## Acknowledgements

This work was funded by NIH grants R01MH106842, R01AG057422, DP2HG010099, R01HG012790, and R01GM122924; a grant from the Knut and Alice Wallenberg Foundation to SciLifeLab for research in Data-driven Life Science, DDLS (KAW 2020.0239); funding from the European Research Council (ERC) under the European Union's Horizon 2020 research and innovation programme (Grant agreement No. 101043238); a European Molecular Biology Organization Postdoctoral Fellowship (ALTF 345–2021) to JD; a Canadian Institutes of Health Research Banting Postdoctoral Fellowship and NIH/NHGRI (K99HG012792) to JAM.

## Additional information

### Competing interests

Júlia Domingo: is CEO and co-founder with equity in Allostery Exploration Technologies, SL. Neville E Sanjana: is an adviser to Qiagen and a co-founder and adviser of TruEdit Bio and OverT Bio. Tuuli Lappalainen: is an advisor and has equity in Variant Bio, was a paid advisor to GSK and has received speaker honoraria from Abbvie and Merck. The other authors declare that no competing interests exist.

### Funding

| Funder | Grant reference number | Author |
|---|---|---|
| National Institute of Mental Health | R01MH106842 | Tuuli Lappalainen |
| National Institute on Aging | R01AG057422 | Tuuli Lappalainen |
| National Human Genome Research Institute | DP2HG010099 | Neville E Sanjana |
| National Human Genome Research Institute | R01HG012790 | Neville E Sanjana Tuuli Lappalainen |
| National Institute of General Medical Sciences | R01GM122924 | Tuuli Lappalainen |
| Knut and Alice Wallenberg Foundation | KAW 2020.0239 | Tuuli Lappalainen |
| European Research Council | 10.3030/101043238 | Tuuli Lappalainen |
| European Molecular Biology Organization | ALTF 345-2021 | Júlia Domingo |
| Canadian Institutes of Health Research | | John A Morris |
| National Human Genome Research Institute | K99HG012792 | John A Morris |

The funders had no role in study design, data collection and interpretation, or the decision to submit the work for publication.

### Author contributions

Júlia Domingo, Conceptualization, Formal analysis, Investigation, Visualization, Methodology, Writing – original draft, Writing – review and editing; Mariia Minaeva, Data curation, Formal analysis, Investigation; John A Morris, Resources, Methodology; Samuel Ghatan, Data curation, Formal analysis, Investigation, Writing – review and editing; Marcello Ziosi, Investigation, Methodology; Neville E Sanjana, Resources; Tuuli Lappalainen, Conceptualization, Supervision, Writing – original draft, Project administration, Writing – review and editing

### Author ORCIDs

Júlia Domingo ⓘ https://orcid.org/0000-0001-6359-2785

Mariia Minaeva https://orcid.org/0000-0001-7028-1264
John A Morris https://orcid.org/0000-0003-2769-8202
Neville E Sanjana https://orcid.org/0000-0002-1504-0027
Tuuli Lappalainen https://orcid.org/0000-0002-7746-8109

Reviewer #1 (Public review): https://doi.org/10.7554/eLife.100555.3.sa1
Reviewer #2 (Public review): https://doi.org/10.7554/eLife.100555.3.sa2
Author response https://doi.org/10.7554/eLife.100555.3.sa3

## Additional files

### Supplementary files
MDAR checklist

### Data availability
All code used in this study is available at GitHub (copy archived at *Minaeva and Domingo, 2026*). Raw sequencing data has been submitted to GEO (accession number GSE257547).

The following dataset was generated:

| Author(s) | Year | Dataset title | Dataset URL | Database and Identifier |
|---|---|---|---|---|
| Domingo J, Minaeva M, Morris JA, Ziosi M, Sanjana NE, Lappalainen T | 2024 | Non-linear transcriptional responses to gradual modulation of transcription factor dosage | https://www.ncbi.nlm.nih.gov/geo/query/acc.cgi?acc=GSE257547 | NCBI Gene Expression Omnibus, GSE257547 |

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
