## [Editor Report · eLife Assessment]

This **important** work develops a new protocol to experimentally perturb target genes across a quantitative range of expression levels in cell lines. The evidence supporting their new perturbation approach is **convincing**, and we propose that focusing on single modality (activation or inhibition) would be sufficient to draw their conclusions. The study will be of broad interest to scientists in the fields of functional genomics and biotechnology.

---

## [Referee Report · Reviewer #1 (Public review)]

In this manuscript, Domingo et al. present a novel perturbation-based approach to experimentally modulate the dosage of genes in cell lines. Their approach is capable of gradually increasing and decreasing gene expression. The authors then use their approach to perturb three key transcription factors and measure the downstream effects on gene expression. Their analysis of the dosage response curve of downstream genes reveals marked non-linearity.

One of the strengths of this study is that many of the perturbations fall within the physiological range for each cis gene. This range is presumably between a single-copy state of heterozygous loss-of-function (log fold change of -1) and a three-copy state (log fold change of ~0.6). This is in contrast with CRISPRi or CRISPRa studies that attempt to maximize the effect of the perturbation, which may result in downstream effects that are not representative of physiological responses.

Another strength of the study is that various points along the dosage-response curve were assayed for each perturbed gene. This allowed the authors to effectively characterize the degree of linearity and monotonicity of each dosage-response relationship. Ultimately, the study revealed that many of these relationships are non-linear, and that the response to activation can be dramatically different than the response to inhibition.

To test their ability to gradually modulate dosage, the authors chose to measure three transcription factors and around 80 known downstream targets. As the authors themselves point out in their discussion about MYB, this biased sample of genes makes it unclear how this approach would generalize genome-wide. In addition, the data generated from this small sample of genes may not represent genome-wide patterns of dosage response. Nevertheless, this unique data set and approach represents a first step in understanding dosage-response relationships between genes.

Another point of general concern in such screens is the use of the immortalized K562 cell line. It is unclear how the biology of these cell lines translates to the in vivo biology of primary cells. However, the authors do follow up with cell-type-specific analyses (Figures 4B, 4C, and 5A) to draw correspondence between their perturbation results and the relevant biology in primary cells and complex diseases.

The conclusions of the study are generally well supported with statistical analysis throughout the manuscript. As an example, the authors utilize well-known model selection methods to identify when there was evidence for non-linear dosage response relationships.

Gradual modulation of gene dosage is a useful approach to model physiological variation in dosage. Experimental perturbation screens that use CRISPR inhibition or activation often use guide RNAs targeting the transcription start site to maximize their effect on gene expression. Generating a physiological range of variation will allow others to better model physiological conditions.

There is broad interest in the field to identify gene regulatory networks using experimental perturbation approaches. The data from this study provides a good resource for such analytical approaches, especially since both inhibition and activation were tested. In addition, these data provide a nuanced, continuous representation of the relationship between effectors and downstream targets, which may play a role in the development of more rigorous regulatory networks.

Human geneticists often focus on loss-of-function variants, which represent natural knock-down experiments, to determine the role of a gene in the biology of a trait. This study demonstrates that dosage response relationships are often non-linear, meaning that the effect of a loss-of-function variant may not necessarily carry information about increases in gene dosage. For the field, this implies that others should continue to focus on both inhibition and activation to fully characterize the relationship between gene and trait.

Comments on revisions:

Thank you for responding to our comments. We have no further comments for the authors.

---

## [Referee Report · Reviewer #2 (Public review)]

Summary:

This work investigates transcriptional responses to varying levels of transcription factors (TFs). The authors aim for gradual up- and down-regulation of three transcription factors GFI1B, NFE2 and MYB in K562 cells, by using a CRISPRa- and a CRISPRi line, together with sgRNAs of varying potency. Targeted single-cell RNA sequencing is then used to measure gene expression of a set of 90 genes, which were previously shown to be downstream of GFI1B and NFE2 regulation. This is followed by an extensive computational analysis of the scRNA-seq dataset. By grouping cells with the same perturbations, the authors can obtain groups of cells with varying average TF expression levels. The achieved perturbations are generally subtle, not reaching half or double doses for most samples, and up-regulation is generally weak below 1.5-fold in most cases. Even in this small range, many target genes exhibit a non-linear response. Since this is rather unexpected, it is crucial to rule out technical reasons for these observations.

Strengths:

The work showcases how a single dataset of CRISPRi/a perturbations with scRNA-seq readout and an extended computational analysis can be used to estimate transcriptome dose-responses, a general approach that likely can be built upon in the future.

Moreover, the authors highlight tiling of sgRNAs +/-1000bp around TSS as a useful approach. Compared with conventional direct TSS-targeting (+/- 200 bp), the larger sequence window allows placing more sgRNAs. Also it requires little prior knowledge of CREs, and avoids using "attenuated" sgRNAs which would require specialized sgRNA design.

Weaknesses:

The experiment was performed in a single replicate and it would have been reassuring to see an independent validation of the main findings, for example through measuring individual dose-response curves .

Much of the analysis depends on the estimation of log-fold changes between groups of single cells with non-targeting controls and those carrying a guide RNA driving a specific knockdown. Generally, biological replicates are recommended for differential gene expression testing (Squair et al. 2021, https://doi.org/10.1038/s41467-021-25960-2). When using the FindMarkers function from the Seurat package, the authors divert from the recommendations for pseudo-bulk analysis to aggregate the raw counts (https://satijalab.org/seurat/articles/de_vignette.html). Furthermore, differential gene expression analysis of scRNA-seq data can suffer from mis-estimations (Nguyen et al. 2023, https://doi.org/10.1038/s41467-023-37126-3), and different computational tools or versions can affect these estimates strongly (Pullin et al. 2024, https://doi.org/10.1186/s13059-024-03183-0 and Rich et al. 2024, https://doi.org/10.1101/2024.04.04.588111). Therefore it would be important to describe more precisely in the Methods how this analysis was performed, any deviations from default parameters, package versions, and at which point which values were aggregated to form "pseudobulk" samples.

Two different cell lines are used to construct dose-response curves, where a CRISPRi line allows gene down-regulation and the CRISPRa line allows gene upregulation. Although both lines are derived from the same parental line (K562) the expression analysis of Tet2, which is absent in the CRISPRi line, but expressed in the CRISPRa line (Fig. S1F, S3A) suggests clonal differences between the two lines. Similarly, the UMAP in S3C and the PCA in S4A suggest batch effects between the two lines. These might confound this analysis, even though all fold changes are calculated relative to the baseline expression in the respective cell line (NTC cells). Combining log2-fold changes from the two cell lines with different baseline expression into a single curve (e.g. Fig. 3) remains misleading, because different data points could be normalized to different base line expression levels.

The study estimates the relationship between TF dose and target gene expression. This requires a system that allows quantitative changes in TF expression. The data provided does not convincingly show that this condition is met, which however is an essential prerequisite for the presented conclusions. Specifically, the data shown in Fig. S3A shows that upon stronger knock-down, a subpopulation of cells appear, where the targeted TF is not detected any more (drop-outs). Also in Fig. 3B (top) suggests that the knock-down is either subtle (similar to NTCs) or strong, but intermediate knock-down (log2-FC of 0.5-1) does not occur. Although the authors argue that this is a technical effect of the scRNA-seq protocol, it is also possible that this represents a binary behavior of the CRISPRi system. Previous work has shown that CRISPRi systems with the KRAB domain largely result in binary repression and not in gradual down-regulation as suggested in this study (Bintu et al. 2016 (https://doi.org/10.1126/science.aab2956), Noviello et al. 2023 (https://doi.org/10.1038/s41467-023-38909-4)).

One of the major conclusions of the study is that non-linear behavior is common. It would be helpful to show that this observation does not arise from the technical concerns described in the previous points. This could be done for instance with independent experimental validations.

Did the authors achieve their aims? Do the results support the conclusions?:

Some of the most important conclusions, such as the claim that non-linear responses are common, are not well supported because they rely on accurately determining the quantitative responses of trans genes, which suffers from the previously mentioned concerns.

Discussion of the likely impact of the work on the field, and the utility of the methods and data to the community:

Together with other recent publications, this work emphasizes the need to study transcription factor function with quantitative perturbations. The computational code repository contains all the valuable code with inline comments, but would have benefited from a readme file explaining the repository structure, package versions, and instructions to reproduce the analyses, including which input files or directory structure would be needed.

---

## [Author Response]

The following is the authors’ response to the original reviews.

**Reviewer #1 (Public review):**
In this manuscript, Domingo et al. present a novel perturbation-based approach to experimentally modulate the dosage of genes in cell lines. Their approach is capable of gradually increasing and decreasing gene expression. The authors then use their approach to perturb three key transcription factors and measure the downstream effects on gene expression. Their analysis of the dosage response curve of downstream genes reveals marked non-linearity.One of the strengths of this study is that many of the perturbations fall within the physiological range for each cis gene. This range is presumably between a single-copy state of heterozygous loss-of-function (log fold change of -1) and a three-copy state (log fold change of ~0.6). This is in contrast with CRISPRi or CRISPRa studies that attempt to maximize the effect of the perturbation, which may result in downstream effects that are not representative of physiological responses.Another strength of the study is that various points along the dosage-response curve were assayed for each perturbed gene. This allowed the authors to effectively characterize the degree of linearity and monotonicity of each dosage-response relationship. Ultimately, the study revealed that many of these relationships are non-linear, and that the response to activation can be dramatically different than the response to inhibition.To test their ability to gradually modulate dosage, the authors chose to measure three transcription factors and around 80 known downstream targets. As the authors themselves point out in their discussion about MYB, this biased sample of genes makes it unclear how this approach would generalize genome-wide. In addition, the data generated from this small sample of genes may not represent genome-wide patterns of dosage response. Nevertheless, this unique data set and approach represents a first step in understanding dosage-response relationships between genes.Another point of general concern in such screens is the use of the immortalized K562 cell line. It is unclear how the biology of these cell lines translates to the in vivo biology of primary cells. However, the authors do follow up with cell-type-specific analyses (Figures 4B, 4C, and 5A) to draw a correspondence between their perturbation results and the relevant biology in primary cells and complex diseases.The conclusions of the study are generally well supported with statistical analysis throughout the manuscript. As an example, the authors utilize well-known model selection methods to identify when there was evidence for non-linear dosage response relationships.Gradual modulation of gene dosage is a useful approach to model physiological variation in dosage. Experimental perturbation screens that use CRISPR inhibition or activation often use guide RNAs targeting the transcription start site to maximize their effect on gene expression. Generating a physiological range of variation will allow others to better model physiological conditions.There is broad interest in the field to identify gene regulatory networks using experimental perturbation approaches. The data from this study provides a good resource for such analytical approaches, especially since both inhibition and activation were tested. In addition, these data provide a nuanced, continuous representation of the relationship between effectors and downstream targets, which may play a role in the development of more rigorous regulatory networks.Human geneticists often focus on loss-of-function variants, which represent natural knock-down experiments, to determine the role of a gene in the biology of a trait. This study demonstrates that dosage response relationships are often non-linear, meaning that the effect of a loss-of-function variant may not necessarily carry information about increases in gene dosage. For the field, this implies that others should continue to focus on both inhibition and activation to fully characterize the relationship between gene and trait.

We thank the reviewer for their thoughtful and thorough evaluation of our study. We appreciate their recognition of the strengths of our approach, particularly the ability to modulate gene dosage within a physiological range and to capture non-linear dosage-response relationships. We also agree with the reviewer’s points regarding the limitations of gene selection and the use of K562 cells, and we are encouraged that the reviewer found our follow-up analyses and statistical framework to be well-supported. We believe this work provides a valuable foundation for future genome-wide applications and more physiologically relevant perturbation studies.

**Reviewer #2 (Public review):**
Summary:This work investigates transcriptional responses to varying levels of transcription factors (TFs). The authors aim for gradual up- and down-regulation of three transcription factors GFI1B, NFE2, and MYB in K562 cells, by using a CRISPRa- and a CRISPRi line, together with sgRNAs of varying potency. Targeted single-cell RNA sequencing is then used to measure gene expression of a set of 90 genes, which were previously shown to be downstream of GFI1B and NFE2 regulation. This is followed by an extensive computational analysis of the scRNA-seq dataset. By grouping cells with the same perturbations, the authors can obtain groups of cells with varying average TF expression levels. The achieved perturbations are generally subtle, not reaching half or double doses for most samples, and up-regulation is generally weak below 1.5-fold in most cases. Even in this small range, many target genes exhibit a non-linear response. Since this is rather unexpected, it is crucial to rule out technical reasons for these observations.

We thank the reviewer for their detailed and thoughtful assessment of our work. We are encouraged by their recognition of the strengths of our study, including the value of quantitative CRISPR-based perturbation coupled with single-cell transcriptomics, and its potential to inform gene regulatory network inference. Below, we address each of the concerns raised:

Strengths:The work showcases how a single dataset of CRISPRi/a perturbations with scRNA-seq readout and an extended computational analysis can be used to estimate transcriptome dose responses, a general approach that likely can be built upon in the future.Weaknesses:(1) The experiment was only performed in a single replicate. In the absence of an independent validation of the main findings, the robustness of the observations remains unclear.

We acknowledge that our study was performed in a single pooled experiment. While additional replicates would certainly strengthen the findings, in high-throughput single-cell CRISPR screens, individual cells with the same perturbation serve as effective internal replicates. This is a common practice in the field. Nevertheless, we agree that biological replicates would help control for broader technical or environmental effects.

(2) The analysis is based on the calculation of log-fold changes between groups of single cells with non-targeting controls and those carrying a guide RNA driving a specific knockdown. How the fold changes were calculated exactly remains unclear, since it is only stated that the FindMarkers function from the Seurat package was used, which is likely not optimal for quantitative estimates. Furthermore, differential gene expression analysis of scRNA-seq data can suffer from data distortion and mis-estimations (Heumos et al. 2023 (https://doi.org/10.1038/s41576-023-00586-w), Nguyen et al. 2023 (https://doi.org/10.1038/s41467-023-37126-3)). In general, the pseudo-bulk approach used is suitable, but the correct treatment of drop-outs in the scRNA-seq analysis is essential.

We thank the reviewer for highlighting recent concerns in the field. A study benchmarking association testing methods for perturb-seq data found that among existing methods, Seurat’s FindMarkers function performed the best (T. Barry et al. 2024).

In the revised Methods, we now specify the formula used to calculate fold change and clarify that the estimates are derived from the Wilcoxon test implemented in Seurat’s FindMarkers function. We also employed pseudo-bulk grouping to mitigate single-cell noise and dropout effects.

(3) Two different cell lines are used to construct dose-response curves, where a CRISPRi line allows gene down-regulation and the CRISPRa line allows gene upregulation. Although both lines are derived from the same parental line (K562) the expression analysis of Tet2, which is absent in the CRISPRi line, but expressed in the CRISPRa line (Figure S3A) suggests substantial clonal differences between the two lines. Similarly, the PCA in S4A suggests strong batch effects between the two lines. These might confound this analysis.

We agree that baseline differences between CRISPRi and CRISPRa lines could introduce confounding effects if not appropriately controlled for. We emphasize that all comparisons are made as fold changes relative to non-targeting control (NTC) cells within each line, thereby controlling for batch- and clone-specific baseline expression. See figures S4A and S4B.

(4) The study uses pseudo-bulk analysis to estimate the relationship between TF dose and target gene expression. This requires a system that allows quantitative changes in TF expression. The data provided does not convincingly show that this condition is met, which however is an essential prerequisite for the presented conclusions. Specifically, the data shown in Figure S3A shows that upon stronger knock-down, a subpopulation of cells appears, where the targeted TF is not detected anymore (drop-outs). Also Figure 3B (top) suggests that the knock-down is either subtle (similar to NTCs) or strong, but intermediate knock-down (log2-FC of 0.5-1) does not occur. Although the authors argue that this is a technical effect of the scRNA-seq protocol, it is also possible that this represents a binary behavior of the CRISPRi system. Previous work has shown that CRISPRi systems with the KRAB domain largely result in binary repression and not in gradual down-regulation as suggested in this study (Bintu et al. 2016 (https://doi.org/10.1126/science.aab2956), Noviello et al. 2023 (https://doi.org/10.1038/s41467-023-38909-4)).

Figure S3A shows normalized expression values, not fold changes. A pseudobulk approach reduces single-cell noise and dropout effects. To test whether dropout events reflect true binary repression or technical effects, we compared trans-effects across cells with zero versus low-but-detectable target gene expression (Figure S3B). These effects were highly concordant, supporting the interpretation that dropout is largely technical in origin. We agree that KRAB-based repression can exhibit binary behavior in some contexts, but our data suggest that cells with intermediate repression exist and are biologically meaningful. In ongoing unpublished work, we pursue further analysis of these data at the single cell level, and show that for nearly all guides the dosage effects are indeed gradual rather than driven by binary effects across cells.

(5) One of the major conclusions of the study is that non-linear behavior is common. This is not surprising for gene up-regulation, since gene expression will reach a plateau at some point, but it is surprising to be observed for many genes upon TF down-regulation. Specifically, here the target gene responds to a small reduction of TF dose but shows the same response to a stronger knock-down. It would be essential to show that his observation does not arise from the technical concerns described in the previous point and it would require independent experimental validations.

This phenomenon—where relatively small changes in cis gene dosage can exceed the magnitude of cis gene perturbations—is not unique to our study. This also makes biological sense, since transcription factors are known to be highly dosage sensitive and generally show a smaller range of variation than many other genes (that are regulated by TFs). Empirically, these effects have been observed in previous CRISPR perturbation screens conducted in K562 cells, including those by Morris et al. (2023), Gasperini et al. (2019), and Replogle et al. (2022), to name but a few studies that our lab has personally examined the data of.

(6) One of the conclusions of the study is that guide tiling is superior to other methods such as sgRNA mismatches. However, the comparison is unfair, since different numbers of guides are used in the different approaches. Relatedly, the authors point out that tiling sometimes surpassed the effects of TSS-targeting sgRNAs, however, this was the least fair comparison (2 TSS vs 10 tiling guides) and additionally depends on the accurate annotation of TSS in the relevant cell line.

We do not draw this conclusion simply from observing the range achieved but from a more holistic observation. We would like to clarify that the number of sgRNAs used in each approach is proportional to the number of base pairs that can be targeted in each region: while the TSS-targeting strategy is typically constrained to a small window of a few dozen base pairs, tiling covers multiple kilobases upstream and downstream, resulting in more guides by design rather than by experimental bias. The guides with mismatches do not have a great performance for gradual upregulation.

We would also like to point out that the observation that the strongest effects can arise from regions outside the annotated TSS is not unique to our study and has been demonstrated in prior work (referenced in the text).

To address this concern, we have revised the text to clarify that we do not consider guide tiling to be inherently superior to other approaches such as sgRNA mismatches. Rather, we now describe tiling as a practical and straightforward strategy to obtain a wide range of gene dosage effects without requiring prior knowledge beyond the approximate location of the TSS. We believe this rephrasing more accurately reflects the intent and scope of our comparison.

(7) Did the authors achieve their aims? Do the results support the conclusions?: Some of the most important conclusions are not well supported because they rely on accurately determining the quantitative responses of trans genes, which suffers from the previously mentioned concerns.

We appreciate the reviewer’s concern, but we would have wished for a more detailed characterization of which conclusions are not supported, given that we believe our approach actually accounts for the major concerns raised above. We believe that the observation of non-linear effects is a robust conclusion that is also consistent with known biology, with this paper introducing new ways to analyze this phenomenon.

(8) Discussion of the likely impact of the work on the field, and the utility of the methods and data to the community:

Together with other recent publications, this work emphasizes the need to study transcription factor function with quantitative perturbations. Missing documentation of the computational code repository reduces the utility of the methods and data significantly.

Documentation is included as inline comments within the R code files to guide users through the analysis workflow.

**Reviewer #1 (Recommendations for the authors):**
In Figure 3C (and similar plots of dosage response curves throughout the manuscript), we initially misinterpreted the plots because we assumed that the zero log fold change on the horizontal axis was in the middle of the plot. This gives the incorrect interpretation that the trans genes are insensitive to loss of GFI1B in Figure 3C, for instance. We think it may be helpful to add a line to mark the zero log fold change point, as was done in Figure 3A.

We thank the reviewer for this helpful suggestion. To improve clarity, we have added a vertical line marking the zero log fold change point in Figure 3C and all similar dosage-response plots. We agree this makes the plots easier to interpret at a glance.

Similarly, for heatmaps in the style of Figure 3B, it may be nice to have a column for the non-targeting controls, which should be a white column between the perturbations that increase versus decrease GFI1B.

We appreciate the suggestion. However, because all perturbation effects are computed relative to the non-targeting control (NTC) cells, explicitly including a separate column for NTC in the heatmap would add limited interpretive value and could unnecessarily clutter the figure. For clarity, we have emphasized in the figure legend that the fold changes are relative to the NTC baseline.

We found it challenging to assess the degree of uncertainty in the estimation of log fold changes throughout the paper. For example, the authors state the following on line 190: "We observed substantial differences in the effects of the same guide on the CRISPRi and CRISPRa backgrounds, with no significant correlation between cis gene fold-changes." This claim was challenging to assess because there are no horizontal or vertical error bars on any of the points in Figure 2A. If the log fold change estimates are very noisy, the data could be consistent with noisy observations of a correlated underlying process. Similarly, to our understanding, the dosage response curves are fit assuming that the cis log fold changes are fixed. If there is excessive noise in the estimation of these log fold changes, it may bias the estimated curves. It may be helpful to give an idea of the amount of estimation error in the cis log fold changes.

We agree that assessing the uncertainty in log fold change estimates is important for interpreting both the lack of correlation between CRISPRi and CRISPRa effects (Figure 2A) and the robustness of the dosage-response modeling.

In response, we have now updated Figure 2A to include both vertical and horizontal error bars, representing the standard errors of the log2 fold-change estimates for each guide in the CRISPRi and CRISPRa conditions. These error estimates were computed based on the differential expression analysis performed using the FindMarkers function in Seurat, which models gene expression differences between perturbed and control cells. We also now clarify this in the figure legend and methods.

The authors mention hierarchical clustering on line 313, which identified six clusters. Although a dendrogram is provided, these clusters are not displayed in Figure 4A. We recommend displaying these clusters alongside the dendrogram.

We have added colored bars indicating the clusters to improve the clarity. Thank you for the suggestion.

In Figures 4B and 4C, it was not immediately clear what some of the gene annotations meant. For example, neither the text nor the figure legend discusses what "WBCs", "Platelets", "RBCs", or "Reticulocytes" mean. It would be helpful to include this somewhere other than only the methods to make the figure more clear.

To improve clarity, we have updated the figure legends for Figures 4B and 4C to explicitly define these abbreviations.

We struggled to interpret Figure 4E. Although the authors focus on the association of MYB with pHaplo, we would have appreciated some general discussion about the pattern of associations seen in the figure and what the authors expected to observe.

We have changed the paragraph to add more exposition and clarification:

“The link between selective constraint and response properties is most apparent in the MYB trans network. Specifically, the probability of haploinsufficiency (pHaplo) shows a significant negative correlation with the dynamic range of transcriptional responses (Figure 4G): genes under stronger constraint (higher pHaplo) display smaller dynamic ranges, indicating that dosage-sensitive genes are more tightly buffered against changes in MYB levels. This pattern was not reproduced in the other trans networks (Figure 4E)”.

Line 71: potentially incorrect use of "rending" and incorrect sentence grammar.

Fixed

Line 123: "co-expression correlation across co-expression clusters" - authors may not have intended to use "co-expression" twice.

Original sentence was correct.

Line 246: "correlations" is used twice in "correlations gene-specific correlations."

Fixed.

**Reviewer #2 (Recommendations for the authors):**
(1) To show that the approach indeed allows gradual down-regulation it would be important to quantify the know-down strength with a single-cell readout for a subset of sgRNAs individually (e.g. flowfish/protein staining flow cytometry).

We agree that single-cell validation of knockdown strength using orthogonal approaches such as flowFISH or protein staining would provide additional support. However, such experiments fall outside the scope of the current study and are not feasible at this stage. We note that the observed transcriptomic changes and dosage responses across multiple perturbations are consistent with effective and graded modulation of gene expression.

(2) Similarly, an independent validation of the observed dose-response relationships, e.g. with individual sgRNAs, can be helpful to support the conclusions about non-linear responses.

Fig. S4C includes replication of trans-effects for a handful of guides used both in this study and in Morris et al. While further orthogonal validation of dose-response relationships would be valuable, such extensive additional work is not currently feasible within the scope of this study. Nonetheless, the high degree of replication in Fig. S4C as well as consistency of patterns observed across multiple sgRNAs and target genes provides strong support for the conclusions drawn from our high-throughput screen.

(3) The calculation of the log2 fold changes should be documented more precisely. To perform a pseudo-bulk analysis, the raw UMI counts should be summed up in each group (NTC, individual targeting sgRNAs), including zero counts, then the data should be normalized and the fold change should be calculated. The DESeq package for example would be useful here.

We have updated the methods in the manuscript to provide more exposition of how the logFC was calculated:

“In our differential expression (DE) analysis, we used Seurat’s FindMarkers() function, which computes the log fold change as the difference between the average normalized gene expression in each group on the natural log scale:

Logfc = log_e(mean(expression in group 1)) - log_e(mean(expression in group 2))

This is calculated in pseudobulk where cells with the same sgRNA are grouped together and the mean expression is compared to the mean expression of cells harbouring NTC guides. To calculate per-gene differential expression p-value between the two cell groups (cells with sgRNA vs cells with NTC), Wilcoxon Rank-Sum test was used”.

(4) A more careful characterization of the cell lines used would be helpful. First, it would be useful to include the quality controls performed when the clonal lines were selected, in the manuscript. Moreover, a transcriptome analysis in comparison to the parental cell line could be performed to show that the cell lines are comparable. In addition, it could be helpful to perform the analysis of the samples separately to see how many of the response behaviors would still be observed.

Details of the quality control steps used during the selection of the CRISPRa clonal line are already included in the Methods section, and Fig. S4A shows the transcriptome comparison of CRISPRi and CRISPRa lines also for non-targeting guides. Regarding the transcriptomic comparison with the parental cell line, we agree that such an analysis would be informative; however, this would require additional experiments that are not feasible within the scope of the current study. Finally, while analyzing the samples separately could provide further insight into response heterogeneity, we focused on identifying robust patterns across perturbations that are reproducible in our pooled screening framework. We believe these aggregate analyses capture the major response behaviors and support the conclusions drawn.

(5) In general we were surprised to see such strong responses in some of the trans genes, in some cases exceeding the fold changes of the cis gene perturbation more than 2x, even at the relatively modest cis gene perturbations (Figures S5-S8). How can this be explained?

This phenomenon—where trans gene responses can exceed the magnitude of cis gene perturbations—is not unique to our study. Similar effects have been observed in previous CRISPR perturbation screens conducted in K562 cells, including those by Morris et al. (2023), Gasperini et al. (2019), and Replogle et al. (2022).

Several factors may contribute to this pattern. One possibility is that certain trans genes are highly sensitive to transcription factor dosage, and therefore exhibit amplified expression changes in response to relatively modest upstream perturbations. Transcription factors are known to be highly dosage sensitive and generally show a smaller range of variation than many other genes (that are regulated by TFs). Mechanistically, this may involve non-linear signal propagation through regulatory networks, in which intermediate regulators or feedback loops amplify the downstream transcriptional response. While our dataset cannot fully disentangle these indirect effects, the consistency of this observation across multiple studies suggests it is a common feature of transcriptional regulation in K562 cells.

(6) In the analysis shown in Figure S3B, the correlation between cells with zero count and >0 counts for the cis gene is calculated. For comparison, this analysis should also show the correlation between the cells with similar cis-gene expression and between truly different populations (e.g. NTC vs strong sgRNA).

The intent of Figure S3B was not to compare biologically distinct populations or perform differential expression analyses—which we have already conducted and reported elsewhere in the manuscript—but rather to assess whether fold change estimates could be biased by differences in the baseline expression of the target gene across individual cells. Specifically, we sought to determine whether cells with zero versus non-zero expression (as can result from dropouts or binary on/off repression from the KRAB-based CRISPRi system) exhibit systematic differences that could distort fold change estimation. As such, the comparisons suggested by the reviewer do not directly relate to the goal of the analysis which Figure S3B was intended to show.

(7) It is unclear why the correlation between different lanes is assessed as quality control metrics in Figure S1C. This does not substitute for replicates.

The intent of Figure S1C was not to serve as a general quality control metric, but rather to illustrate that the targeted transcript capture approach yielded consistent and specific signal across lanes. We acknowledge that this may have been unclear and have revised the relevant sentence in the text to avoid misinterpretation.

“We used the protein hashes and the dCas9 cDNA (indicating the presence or absence of the KRAB domain) to demultiplex and determine the cell line—CRISPRi or CRISPRa. Cells containing a single sgRNA were identified using a Gaussian mixture model (see Methods). Standard quality control procedures were applied to the scRNA-seq data (see Methods). To confirm that the targeted transcript capture approach worked as intended, we assessed concordance across capture lanes (Figure S1C)”.

(8) Figures and legends often miss important information. Figure 3B and S5-S8: what do the transparent bars represent? Figure S1A: color bar label missing. Figure S4D: what are the lines?, Figure S9A: what is the red line? In Figure S8 some of the fitted curves do not overlap with the data points, e.g. PKM. Fig. 2C: why are there more than 96 guide RNAs (see y-axis)?

We have addressed each point as follows:

Figure 3B: The figure legend has been updated to clarify the meaning of the transparent bars.

Figures S5–S8: There are no transparent bars in these figures; we confirmed this in the source plots.

Figure S1A: The color bar label is already described in the figure legend, but we have reformulated the caption text to make this clearer.

Figure S4D: The dashed line represents a linear regression between the x and y variables. The figure caption has been updated accordingly.

Figure S9A: We clarified that the red line shows the median ∆AIC across all genes and conditions.

Figure S8: We agree that some fitted curves (e.g., PKM) do not closely follow the data points. This reflects high noise in these specific measurements; as noted in the text, TET2 is not expected to exert strong trans effects in this context.

Figure 2C: Thank you for catching this. The y-axis numbers were incorrect because the figure displays the proportion of guides (summing to 100%), not raw counts. We have corrected the y-axis label and updated the numbers in the figure to resolve this inconsistency.

(9) The code is deposited on Github, but documentation is missing.

Documentation is included as inline comments within the R code files to guide users through the analysis workflow.

(10) The methods miss a list of sgRNA target sequences.

We thank the reviewer for this observation. A complete table containing all processed data, including the sequences of the sgRNAs used in this study, is available at the following GEO link:

https://www.ncbi.nlm.nih.gov/geo/download/?acc=GSE257547&format=file&file=GSE257547%5Fd2n%5Fprocessed%5Fdata%2Etxt%2Egz

(11) In some parts, the language could be more specific and/or the readability improved, for example:Line 88: "quantitative landscape".

Changed to “quantitative patterns”.

Lines 88-91: long sentence hard to read.

This complex sentence was broken up into two simpler ones:

“We uncovered quantitative patterns of how gradual changes in transcription dosage lead to linear and non-linear responses in downstream genes. Many downstream genes are associated with rare and complex diseases, with potential effects on cellular phenotypes”.

Line 110: "tiling sgRNAs +/- 1000 bp from the TSS", could maybe be specified by adding that the average distance was around 100 or 110 bps?Lines 244-246: hard to understand.

We struggle to see the issue here and are not sure how it can be reworded.

Lines 339-342: hard to understand.

These sentences have been reworded to provide more clarity.

(12) A number of typos, and errors are found in the manuscript:Line 71: "SOX2" -> "SOX9".

FIXED

Line 73: "rending" -> maybe "raising" or "posing"?

FIXED

Line 157: "biassed".

FIXED

Line 245: "exhibited correlations gene-specific correlations with".

FIXED

Multiple instances, e.g. 261: "transgene" -> "trans gene".

FIXED

Line 332: "not reproduced with among the other".

FIXED

Figure S11: betweenness.

This is the correct spelling

There are more typos that we didn't list here.

We went through the manuscript and corrected all the spelling errors and typos.